# Choosing the right strategies: An analysis of crisis response strategies in Chinese universities

**Yongshi Liu[1]\*, Deyi Gao[2], Hongtao Duan[3]**

1 Business School, University of Shanghai for Science and Technology, Shanghai, China, 2 Shanghai Municipal Government Major Administrative Decision-Making Consulting, Shanghai Municipal Government, Shanghai, China, 3 Shanghai Network Culture Development and Research Center of Education System, Shanghai, China

\* liuyongshi1995@126.com

## Abstract

Situational Crisis Communication Theory (SCCT) provides a theoretical framework for crisis response strategies in different crisis situations. Based on the theoretical framework of SCCT, a qualitative thematic analysis is conducted to examine the crisis response strategies used in 182 response documents officially released by Chinese universities, and a content analysis is further employed to classify 147 crises in 125 universities into three clusters to examine the use of strategies in addressing different types of crises. The findings revealed that Chinese universities have adopted 17 new strategies beyond the strategies proposed by SCCT, highlighting a discrepancy between the theoretical ground of the theory and the application of strategies in real-life crisis response of Chinese universities. Furthermore, the results revealed the applicable strategies for different crisis clusters, which contributed to construct the framework of crisis response strategies for Chinese universities in different situations. This study expanded SCCT theoretically by enriching the crisis response strategies while practically improving the applicability of SCCT in Chinese universities. The findings provided guiding significance for Chinese universities administrators in developing and implementing effective crisis response strategies in real-life crisis situations, enabling them to adjust and optimize their plans by incorporating innovative strategies to enhance the effectiveness of responses. These innovative strategies also provided empirical evidence for other organizations in different cultural contexts in crisis response.

## 1. Introduction

With the development of social media in China, the Chinese public has increasingly gained the right to publish information and opinions [1]. Public's awareness of political participation has been continuously enhanced, and the active degree of public speech and enthusiasm for political participation have reached an unprecedented height [2]. Public opinion crises on social media have frequently occurred in China. In recent years, numerous crises in the fields of public health, food safety, medical malpractice, government management, product quality,

**Data availability statement:** All relevant data are within the manuscript and its Supporting Information files.

**Funding:** The author(s) received no specific funding for this work.

**Competing interests:** The authors have declared that no competing interests exist.

security incidents, corporate image have aroused widespread public concern and discussion. The occurrence of public opinion crisis makes the relevant responsible parties under immense pressure in crises response. When public opinion crises arise, if governments, institutions, and enterprises fail to respond reasonably, it will cause severe damage to their own image, trigger further public opinion crises, and undermine their credibility [3]. Obviously, public opinion crisis response has become an important research topic at present.

Academics have shown great interest in research of these public opinion crises in China, and the relevant research mainly focus on three points. First, researchers deeply discussed the interaction between government crisis management and public opinion, and analyzed the influence of public opinion on government management strategies and how the government effectively responds to these public opinions [4–7]. Second, some researchers are committed to revealing the evolution law of public opinion through big data and computer models, providing valuable insights for formulating effective response strategies [8–11]. Third, many researchers have conducted research on how enterprises and brands respond to negative public opinions on Chinese social media during crises [12–15]. Academics attaches great importance to research on public opinion crisis and crisis response, which not only has profound theoretical value, but also shows its indispensable importance in reality.

In China, under the jurisdiction of the National Government and the Ministry of Education, Chinese universities are important institutions for talent cultivation. There has been an increasing number of crises about Chinese universities on the internet in recent years. The reasons are that, first, compared with other organizations, Chinese universities have a better social image and evaluation, which is easy to cause public opinion crisis when there is incident contrary to their image and public expectations, and second, since the Internet has become an important way for Chinese citizens to supervise the management of universities and the behavior of internal members, Chinese universities are under great pressure in public opinion crisis response. In 2018, Tao, a graduate student at Wuhan University of Technology, committed suicide due to long-term oppression and bullying by his tutor, which forced Wuhan University of Technology to face the doubts of the public and the media and offer reasonable explanations and solutions; In 2019, Zhai, a well-known Chinese actor, was found to have committed academic fraud, causing widespread concern and leading Beijing Film Academy and Peking University to fall into a serious crisis of public opinion. The incident not only affected the fairness of the examination and admission procedures, but also caused great impact and damage to the credibility of the entire Chinese education system, forcing the two universities to explain and apologize to the public.

Previous studies have shown that organizations that reply to public messages may achieve a more positive audience assessment of their crisis communication efforts and their effectiveness in handling crises [16]. Crisis response is of great significance to repair the image of the universities, particularly since social media has become the fastest communication tool between organizations and the public and enables organizations to achieve real-time communication with the public. Chinese universities must respond strategically to the crisis, otherwise it will have a negative impact on the sustainable development of universities. However, despite the prevalence of crises in Chinese universities, there is limited research on how to respond. In particular, empirical studies on crisis response in Chinese universities, are almost nonexistent. To bridge the research gap, this study conducts qualitative thematic analysis of 182 response documents from 147 public opinion crises in 125 universities, aiming to reveal the use of crisis response strategies in Chinese universities, and constructs a framework between crisis response strategies and different situations in Chinese universities based on the classification of crisis clusters through content analysis. In other words, the findings of this study provide a comprehensive and definitive answer to the question that has yet to be

addressed in current research: what strategies are used in the public opinion crisis response in Chinese universities and what strategies are applicable to different types of crises.

## 2. Theory and literature review

### 2.1. Crisis in Chinese universities

Crisis in higher education institutions is occurring all over the world. Crisis not only hurt the reputation of a university but may also strain its relationship with key internal and external stakeholders [17]. However, there is subtle difference between Chinese and western studies on crisis in universities. Western scholars often use terms such as "scandal" and "crisis" to refer to negative incidents in universities, while in Chinese, it is customary to use the word of "Yuqing", which means public opinion crisis. But even if the words are different, the foundation is the same: incidents that affect the reputation of the universities and threaten the safety of faculty and students. Therefore, the crisis of Chinese universities in this study refers to the public opinion crisis.

Public opinion crisis in Chinese universities is the public opinion and attitude towards the events related to higher education institutions [18], which is often sudden or unexpected, that disrupts the normal operations of the institution or its educational mission and threatens the well-being of personnel, property, financial resources, or reputation of the institutions [19], and thus would affect the application for admission of the universities involved and their cultivating effect and their long-term development [20]. Public opinion crises in Chinese universities are omnipresent. From the heinous sexual abuse scandals of university teachers to academic fraud, public opinion crises in Chinese universities have always triggered widespread discussion among the Chinese public and high media interest.

Due to the rapid development of the Internet, negative news of universities reported by the media and the network public opinion generated by social platforms further amplify the harm of public opinion crisis in Chinese universities. Also, students at universities with active thinking, high education, and proficiency in using social media have become the mainstream of Internet users which makes the public opinion crises in universities attract more and more attention from the public and media [21]. The frequency and severity of controversies over teacher behavior, fairness of examination, and campus management have increased, causing universities under enormous pressure to respond to public opinion crises. Therefore, the universities must take the public opinion crisis seriously and make strategic management in order to protect the normal operation and long-term development of the universities.

### 2.2. Crisis response strategies in universities

The distinctive status of universities requires crisis communication research to be adapted to them. As an elite cultivating institution, the public often expects universities to have an "elite" approach in crisis response, and the moral requirements of universities are often higher than other organizations. It is common for enterprises to pursue their own benefit, but universities are generally required to protect the safety and interests of students. The universities have the moral duty to protect their students from the harm causing by public opinion by effective communication in the time of crisis [22]. It is necessary for universities to adopt appropriate strategies to respond to the crises.

Different researchers have conducted many studies on crisis response strategies through different university crisis cases. Tulika M. Varma (2011) [17] examined the crisis response strategies of Louisiana State University in the United States after its women's head basketball coach resigned amid charges of inappropriate conduct with former basketball players and found that the university's crisis response strategies contained denial, shifting of the blame,

evasion of responsibility, minimization of the negative effects through the strategies of bolstering and transcendence. More importantly, the study illustrates the value of negotiations in management of relationships during a university crisis. Melanie Formentin et al. (2017) [23] examined social media communication by Penn State University (PSU) in the United States and corresponding stakeholder responses and found that ingratiation and accidental strategies received positive and supportive responses while strategies of defeasibility, reminder, scapegoat, and provocation all received more negative reactions than expected. Walter BLW et al. (2021) [24]examined the responses of two Canadian universities to sexual assault scandals and found that adjusting strategies by observing stakeholder reactions to response strategies is a crucial part of the crisis response process. In particular, McGill University's strategies of shifting from denial and excuses to corrective action proved to be effective during the sexual assault scandals.

However, most of the research on crisis response in Chinese universities is from macroscopic aspects, such as increasing crisis consciousness, constructing university crisis management group, developing crisis warning system and carrying on effective communication [25]. Specific and empirical studies on the crisis response strategies in Chinese universities for crisis managers are still rare. Different from Western countries, Chinese universities are almost all public universities. The crisis response strategies in universities may be quite different between China and the western countries. Schwaag Serger et al. (2015) [26]suggested that the governance of Chinese universities distinguishes itself from the western model, because its complexity in political and economic grounds. Governmental financial support accounts for a significant part for most of the Chinese public universities. The governance system in Chinese universities is unique from the global perspective, i.e., they are dictated by the government to execute the institutional and national objectives [27]. Therefore, it's necessity of conducting China-specific studies on crisis response strategies in universities.

## 2.3. Situational crisis communication theory

Crisis communication is crucial in organizational communication. No organization is immune from a crisis anywhere in the world, even if that organization is vigilant and actively seeks to prevent crises [28]. In fact, situational crisis communication theory (SCCT) has been widely recognized as a mature theoretical system in the field of crisis communication. The crisis response strategies proposed by this theory have been proved to be effective in both for profit and non-profit organizations. SCCT aims to determine how organizations communicate effectively in times of crisis in order to minimize the damage of crisis to organizations' reputation and it has been used by organizations to strategically respond to crises in order to influence stakeholders' perceptions of crises and organizations in crisis [29].

SCCT established three primary and one secondary crisis response strategies, and there are 10 sub-strategies under these four main strategies: (1) deny (attack the accuser, denial, scapegoat); (2) diminish (excuse, justification); (3) rebuild (compensation, apology) and (4) bolstering (reminder, ingratiation, victimage) [30]. Even though the SCCT provides a realistic and effective guidance for organizations to respond to crises, there are still many scholars have explored crisis response strategies in different fields based on it, which has contributed to the development and expansion of SCCT.

Shin, J.H. (2008) [31]conducted a national survey among religious public relations practitioners who working for mainstream Protestant Christian churches. The study found new strategies used by religious public relations professionals in the conflict and crisis, which were collaborating, contending, compromising, concession and corrective action. Liu (2010) [32] conducted a quantitative content analysis of all available documents (104 in total) released by the parties responding to the five racially charged crises and identified four new strategies

currently not included in SCCT: ignore, separate, transcendence, endorsement. Jin and Liu (2010) [33] proposed a new model which primarily grounded in blog-mediated crisis communication (BMCC) and added new response strategies including ignore, separation, transcendence, endorsement, and legal action. Kim and Liu (2012) [34] investigated how 13 corporate and government organizations responded to the first phase of the 2009 flu pandemic through the collection of 2,240 media materials released by 11 organizations. Their study found that organizations used two strategies which currently are not part of SCCT: enhancing and transferring. Lai and Tang (2018) [35] discovered six new strategies not mentioned by SCCT by analyzing 64 cases of Chinese public opinion crisis, including gratitude, accountability, punishment, explanation, information disclosure and attitude indication. Castonguay and Lowes (2022) [36] analyzed the crisis response strategies used by the National Football League (NFL) in its external communications to address the concussion crisis spanning 2015–2020 through a qualitative thematic analysis. Their research identified two new crisis response strategies: organizational change and sharing responsibility. Tian and Yang (2022) [37] applied SCCT in political crisis communication amidst the COVID-19 outbreak which conducted a systematic content analysis of two politicians' tweets, Trump and Cuomo, to evaluate crisis communication strategies. The results indicated that three strategies categorized by SCCT, deny, diminish, and bolstering, surfaced with significance and cohesion as a new strategy specific to the political context was identified.

By reviewing previous studies, as an evolving crisis communication theory, SCCT provides concrete guidelines for managing different types of crises [32]. This study attempts to conduct the qualitative method of thematic analysis and content analysis, to investigate the following issues by taking Chinese universities as the research object, with the purpose of providing a complete theoretical framework and reference basis for the appropriate response strategies that Chinese universities should apply in the face of different crises. Two main questions of the study are proposed as follows:

RQ1: What crisis response strategies were used by Chinese universities in their external communications to respond to the crises?

RQ2: What is the use of strategies for different crisis clusters in Chinese universities? In which types of crises do different response strategies apply?

## 3. Method

### 3.1. Data collection

In terms of research design, a case study was chosen for this research because case study is often used in risk and crisis research [38]. Case study offers the advantage of inductive assessment which is appropriate in policy or strategy implementation analyses [39], and provide insights into specific actions taken by entities during crises. In this study, representative crises on Chinese social media from August 2016 to August 2023 are selected as the research objects, since the notice of "Further Improving Response Capabilities to Government Affairs Crises in the Work of Openness of Government Affairs" became a landmark in China's crisis response, which was issued in August 2016 by the Chinese government to require clearly that "governments at all levels and their departments should attach great importance to the response to public opinion crisis in government affairs and implement the responsibility of response" [40]. The release of the notice standardized the public opinion crisis response and promoted the public opinion crisis response to become the normalization of the government and various institutions [41]. Since then, the response to the public opinion crisis in Chinese universities has also begun to increase gradually. Therefore, this study selected crises between the time when the notice was released and the end of the study period as the research objects of this

study. The samples were considered from four dimensions: topic sensitivity, event impact, media attention and public discussion, which were selected by combining the hot topics list of Sina Weibo, the most popular microblogging website in China [42], and the public opinion crisis response case-base of the National University Online Public Opinion Monitoring Center.

As the most popular social media platform in China, Sina Weibo holds a hot topic list, in which there are 11 topics with the largest volume of searches displayed on the home page. The list is a real-time hot topic detecting system, which provides an objective ranking on the hotness of topics [43]. The topics on the list represent what people care about most. Therefore, the samples selected through Sina Weibo hot topics are persuasive in terms of media attention and public discussion. Besides, the National University Online Public Opinion Monitoring Center is a department specifically established for the public opinion crisis response in universities which monitors the information on the Chinese internet every day about the online public opinion of universities that may trigger public concern and discussion, alerts crisis managers of universities to pay attention, provides constructive suggestions for them, and includes each year's representative cases into their crisis response case-base. Therefore, the samples selected through the case-base of the National University Online Public Opinion Monitoring Center are persuasive in terms of topic sensitivity and event impact. The cases from its case base provide an effective supplement for the sample selection of this study.

Then, all the response documents released by universities on social media during the occurrence of public opinion crisis were collected. Many studies have confirmed that in order to manage a crisis successfully, an organization should communicate strategically and effectively with its stakeholders by relaying information in an efficient and timely manner. This is the reason why communication platforms such as social media become important strategically. On top of greatly increasing the speed of communication between organizations and audiences [44], social media outlets facilitate real-time dialogue between these groups. As the purpose of this research is to determine how the Chinese universities communicated about crises in their external communications, news articles or interviews of school staff were excluded to avoid the subjectivity of a third party in the reporting and framing of the universities' responses. All documentary evidence is officially released by universities. Meanwhile, due to stakeholders perceptions may be influenced by their own interests, and potential emotional factors and biases towards the organization, in order to focus on the coding of institutionalized strategies in universities' official responses, and reveal what response strategies Chinese universities choose under the institutional logic of organizational legitimacy and policy compliance when responding to crises, while maintaining the objectivity of the research, the influence of stakeholders perceptions are also excluded in this study.

Finally, the database of this study was formed. A total of 147 public opinion crises in Chinese universities were collected, including 182 response documents. Among them, 117 cases released one response document, 27 cases released two response documents, 2 cases released three response documents, and 1 case released five response documents, involving a total of 125 different universities. These public opinion crises cover different types of incidents such as teacher and student ethics, campus management, safety accidents, sexual assault and harassment, exam fairness, academic misconduct, and ethnic issues. After the issuance of the response documents, the popularity of public opinion decreased significantly and faded within a week, and no secondary public opinion crisis occurred that the response documents of Chinese universities in these crises are considered to be initially effective in this study. The research results can be inferred to provide reference information for other universities of which response strategies should use in the crises in the future. Table 1 shows 20 examples of crises.

## 3.2. Research approach

To answer RQ1 about which is aimed to examine what crisis response strategies were used by Chinese universities in their external communications to respond to the crises, the method of qualitative thematic analysis is adopted by this study. Thematic analysis is flexible and compatible with various research paradigms and theoretical approaches, and it is a method for identifying, analyzing, and reporting patterns(themes) within a data. It minimally organizes and describes your data in rich detail [45]. The goal of thematic analysis is to identify themes within qualitative data that can be used to interpret and make sense of the data [46]. This approach allows researchers to examine the underlying motivations, ideas, and assumptions within what has been said. This study followed the six-steps framework proposed by Braun and Clarke (2006) [45]: (a) familiarizing with the data, (b) generating initial codes, (c) generating themes, (d) reviewing potential themes, (e) defining and naming themes, and (f) producing a report. Taking a sentence as a unit, this study summarized the relevant excerpt in short phrase representing a code, and every code provides "rich descriptions of the data in an organized and meaningful way" [47].

In this study, two researchers participated in the coding work. One is a doctoral student in public opinion crisis, and the other is a staff engaged in the work related to public opinion crisis response. In the first step, the strategies proposed in SCCT and the references mentioned

**Table 1. Examples of crisis.**

| Case | University | Crisis | Time | No. of response |
|---|---|---|---|---|
| NO.1 | East China Normal University | Celebrated birthday for international students during closed-off management of COVID-19 pandemic. | 2022.04 | 1 |
| NO.2 | Fudan University | A teacher killed his leader. | 2021.06 | 3 |
| NO.3 | Beijing Institute of Technology | A woman kissed an academician during the live online meeting. | 2022.12 | 2 |
| NO.4 | Lanzhou Jiaotong University | A teacher was laid off because of cancer. | 2016.08 | 2 |
| NO.5 | Communication University of China | Be questioned about gender inequality in enrollment. | 2021.03 | 1 |
| NO.6 | Renmin University of China | Postgraduate exam answers leaked before the test. | 2021.04 | 1 |
| NO.7 | Shanxi University of Finance and Economics | A student died in the PE class. | 2022.03 | 1 |
| NO.8 | Shandong University of Technology | A student abused cats. | 2020.04 | 2 |
| NO.9 | Peking University & Beijing Film Academy | A well-known actor committed academic fraud during his doctoral and postdoctoral studies which damaged the credibility of Chinese education system seriously. | 2019.02 | 5 |
| NO.10 | Wuhan University of Technology | A student who suspected of being bullied by tutor killed himself. | 2018.03 | 2 |
| NO.11 | Southwest Jiao Tong University | A student's father helped her cheat in the postgraduate recommendation exam. | 2020.06 | 3 |
| NO.12 | Xi'an International Studies University | Illegal overcharging of tuition. | 2017.03 | 1 |
| NO.13 | Shanghai Jiao Tong University | A tutor insulted his students. | 2019.03 | 1 |
| NO.14 | University of International Business and Economics | An associate professor sexually harassed a female student for a long time. | 2018.01 | 1 |
| NO.15 | Guilin University of Tourism | Internet rumors about school security guards encouraged students to jump from high building. | 2020.10 | 1 |
| NO.16 | Changchun University of Science and Technology & Hebei Normal University | The woman involved in the incident of "driving car into the Forbidden City" was suspected of cheating in exam. | 2020.01 | 2 |
| NO.17 | Shaanxi University of Technology | Recruited three teachers while two of whom were the children of school leaders. | 2022.05 | 2 |
| NO.18 | Qingdao Binhai University | A student fell from a building and died. | 2021.12 | 1 |
| NO.19 | Tongji University | Unqualified pork was found in the canteen. | 2022.04 | 1 |
| NO.20 | Southeast University | The head of the school's publicity department posted pornographic images in an online working group. | 2023.04 | 1 |

above are sorted out as the original identified codes and themes. The Chinese university response documents are coded in sentences, and potential new strategies that do not fit the predetermined codes are recorded in new codes. Generating codes requires marking interesting features of the data in a systematic way. Then, all the codes are collected and sorted, the strategies that fit the definitions are classified into the predetermined themes, and any potential new themes that do not fit the predetermined themes are temporarily placed in the miscellaneous category which is added to account for any incongruous evidence. The researchers investigate whether the themes have "central organizing concepts". Researchers generate new themes and then name and define the new themes based on the central concepts.

For more clarification, it is significant to verify the transferability, dependability and confirmability of data in qualitative thematic analysis. The concepts of transferability, dependability and confirmability were thoroughly examined and verified in this study. Transferability is determined by achieving reference adequacy by archiving the second half of the dataset until the first half is analyzed, and then comparing the data in the second half with the first half. Dependability is verified by peer debriefing. One researcher provides another with detailed notes showing how he came to his findings, and another researcher provides an external check to ensure that the observations and interpretations of the data are valid. Confirmability is achieved by researchers detailing their notes in a systematic way to illustrate the connections between their data and findings [48]. Through the verification, the coding scheme is basically determined.

In the second step, according to the coding scheme, coders were asked to code all the response documents sentence by sentence, but the same strategy used repeatedly in the same crisis was still recorded as once in the final count. To further ensure the full understanding and consistency of the coding scheme between the two coders, 20% of the documents were conducted preliminary test on request [49]. Before the comprehensive implementation of the coding, any inconsistencies in coding or new discovered response strategies are introduced to a third researcher for further discussion until a consensus was reached.

Also, due to the influence of cultural context on the interpretation of data, it is worth noting that although the explanations and the example sentence for each strategy in this study were translated into English, researchers actually used Chinese during the analysis, because analysis in the original language of the document can fully reveal the use of strategies in the particular context, so as to ensure the accuracy and effectiveness of the analysis results. Meanwhile, we noted that many sentences used multiple strategies, so double coding is allowed in this study, because every piece of evidence relevant to the study's objective of identifying the Chinese universities crisis response strategies had to be accounted.

To answer RQ2 about which is aimed to reveal what is the use of strategies for different crisis clusters in Chinese universities and in which type of crises do different response strategies apply, content analysis is conducted in this study to classify 147 crises into three clusters based on the theoretical framework of SCCT. Content analysis is one of the most important research techniques in social sciences to analyze data with a specific context [50]. Thus, when using content analysis, the researchers should have a broader understanding of the context. Content analysis provides a systematic and objective means to make valid inferences from verbal, visual, or written data to describe and quantify specific phenomena in a conceptual form [51]. There are three main steps in content analysis: (a)obtaining the sense of the whole data, selecting the unit of analysis, (b)creating categories and open coding, (c)reporting the analysis process and the results through conceptual system [52].

In this section, taking a case as a unit, the two coders coded independently to divide Chinese universities crises into three clusters after fully understanding the relevant concepts of crisis classification based on SCCT and the research requirements of classification category.

The victim cluster, the accident cluster and the intentional cluster are the three first-level codes of this coding process. First, 20 cases were randomly selected for independent coding. After coding, consistency judgment is carried out. A reliability of 0.863 is achieved using Krippendorf's alpha, which is higher than the standard of 0.7 [53] and shows that two coders have a high degree of agreement in understanding data, context, and coding concepts in the content analysis. Also, the sample size of preliminary test meets the requirement of 10%-20% of the total samples [49]. All remaining disagreements were completely discussed prior to the next implementation of coding until a consensus was reached. In the second round, 50 cases were randomly selected from the remaining cases for independent coding. The coding with inconsistent results were discussed prior to the next implementation of coding until a consensus was reached. In the third round, the remaining 77 cases were coded independently, and the coding with inconsistent results were discussed until a consensus was reached. The discussion after each round of coding makes the next round of coding less disagreement, and the conceptual system of crisis classification of Chinese universities are basically formed, which provides clues for Chinese universities to classify the crisis clusters quickly and reliably. After three rounds of coding, 147 cases were finally classified by the degree of responsibility.

## 4. Findings and discussion

### 4.1. Crisis response strategies in Chinese universities

SCCT provided the framework that guided the thematic analysis, but we still found many differences in crisis response in Chinese universities. More precisely, 24 subthemes are found in the crisis response documents of Chinese universities in 147 cases which include attack the accuser, denial, scapegoat, excuse, transference, separation, compensation, apology, commitment, remedy, investigation, accountability, punishment, explanation, information disclosure, reminder, endorsement, appeal, warning, valued highly, concern, condolence, gratitude, and statement. There are 3 strategies proposed by SCCT, but they are not found being used by Chinese universities which include justification, ingratiation and victimage. 17 new strategies are found which include transference, separation, commitment, remedy, investigation, accountability, punishment, explanation, information disclosure, endorsement, appeal, warning, valued highly, concern, condolence, gratitude, and statement.

According to previous research mentioned above [33–35] transference and separation are added to the diminish strategy, while commitment and remedy are added to the rebuild strategy of the primary crisis response strategy. Endorsement, appeal, and warning are added to the bolstering strategy of the secondary crisis response strategy. Furthermore, a new theme is added to the primary crisis response strategy, which is considered as condition exposure strategy in this study, and it contains five subthemes of investigation, accountability, punishment, explanation, and information disclosure. A new theme is added to the secondary crisis response strategy, which is considered as attitude indication strategy in this study, and it contains five subthemes of valued highly, concern, condolence, gratitude, and statement. All secondary strategies are best used as supplements to the three primary strategies to boost the communication effect and promote the reputation protection of organizations. But secondary strategies should not be used as standalone strategies, because they play a limited role. Table 2 shows interpretations for each strategy. Real-life examples of each strategy are shown in S1 Table in the section of supporting information. For each example sentence, the corresponding crisis can be found in Table 1.

Although this research is following a specific theoretical framework for crisis response in analysis, it is unlikely that the Chinese universities followed the guidance completely to

**Table 2. Definition and naming for crisis response strategies and the coding scheme.**

| Coding | Strategy | Description |
|---|---|---|
| | *Primary Crisis Response Strategy* | |
| | *Deny* | |
| A1 | Attack the accuser | The university confronts the person or group claiming something is wrong with the university. |
| A2 | Denial | The university asserts that there is no crisis. |
| A3 | Scapegoat | The university blames some person or group outside the university for the crisis. |
| | *Diminish* | |
| B1 | Excuse | The university minimizes university responsibility by denying intent to do harm and/or claiming inability to control the events that triggered the crisis. |
| B2 | Justification# | The university minimizes the perceived damage caused by the crisis. |
| B3 | Transference*[34] | Deny the serious problems and admit the less serious problems of the allegation or change the subject from uncontrollable problems to controllable problems. |
| B4 | Separation*[33] | Claiming that university has nothing to do with the initiator for the crisis. |
| | *Rebuild* | |
| C1 | Compensation | The university offers money or provide other material support to victims. |
| C2 | Apology | The university indicates the university takes full responsibility for the crisis and asks stakeholders for forgiveness. |
| C3 | Commitment* | The university promises to take action to assume responsibility or correct mistakes. |
| C4 | Remedy*[31] | Take substantive actions such as improving regulations or practices to reduce harm or prevent a similar crisis from happening again. |
| | *Condition Exposure* * | |
| D1 | Investigation* | Take action to investigate the causes or the facts of the crisis. |
| D2 | Information disclosure*[35] | Expose the facts about the crisis to the public. |
| D3 | Explanation*[35] | Explain the intent of policies and actions to eliminate the misunderstanding of the accuser. |
| D4 | Accountability*[35] | Hold accountable those within the university who have leadership responsibility for the crisis. |
| D5 | Punishment*[35] | The person responsible for the crisis shall be punished or dealt with by law accordingly. |
| | *Secondary Crisis Response Strategy* | |
| | *Bolstering* | |
| E1 | Reminder | Tell stakeholders about the past good works of the university. |
| E2 | Ingratiation# | The university praises stakeholders and/or reminds them of past good works by the university. |
| E3 | Victimage# | The university reminds stakeholders that the university is a victim of the crisis too. |
| E4 | Endorsement*[33] | Verify the fact or the correctness of the crisis management action through the statement of the third-party authority. |
| E5 | Appeal* | Take the crisis as an opportunity to put forward positive advocacy and appeal to the public to reduce the possibility of crisis recurrence. |
| E6 | Warning* | Warning the public not to make the same mistakes that led to the crisis. |
| | *Attitude Indication* *[35] | |
| F1 | Valued highly* | Demonstrate to the public that the university is taking the crisis seriously. |
| F2 | Concern* | Show concern for those related personnel who have been hurt in the crisis. |

*(Continued)*

**Table 2.** (Continued)

| Coding | Strategy | Description |
|---|---|---|
| F3 | Condolence* | Express condolences or regret to the victims who have lost their lives in the crisis. |
| F4 | Gratitude* [35] | Express gratitude to society, the media, and the public for their attention or scrutiny for the university. |
| F5 | Statement* | Make clear to the public the principles and positions of the university, show the attitude of not conniving at those responsible for the crisis and resolutely safeguarding the public interest. |

*Added strategies to SCCT. #Deleted strategies from SCCT.

respond to crises due to different national conditions and undetermined theoretical applicability. Consequently, it is crucial to remain open-minded for researchers when conducting exploratory research to capture all information relevant to the research objectives. This is exactly the main purpose of this study to reveal the discrepancy between SCCT and the application of crisis response strategies of real-life crises in Chinese universities and extend SCCT theoretically.

## 4.2. The use of response strategies in Chinese universities

The number of response strategies used refers to the number of different response strategies used in a crisis. The same strategy is recorded as one time when it is used repeatedly in the same crisis. After coding, the statistical results show that a total of 601 response strategies were used in 147 crises in Chinese universities, with an average of 4.02 response strategies used in each crisis, indicating that Chinese universities usually use a combination of strategies in crisis response. Fig 1 shows the strategies that Chinese universities use most in crisis response are valued highly, punishment, information disclosure, commitment and investigation, while the strategies used

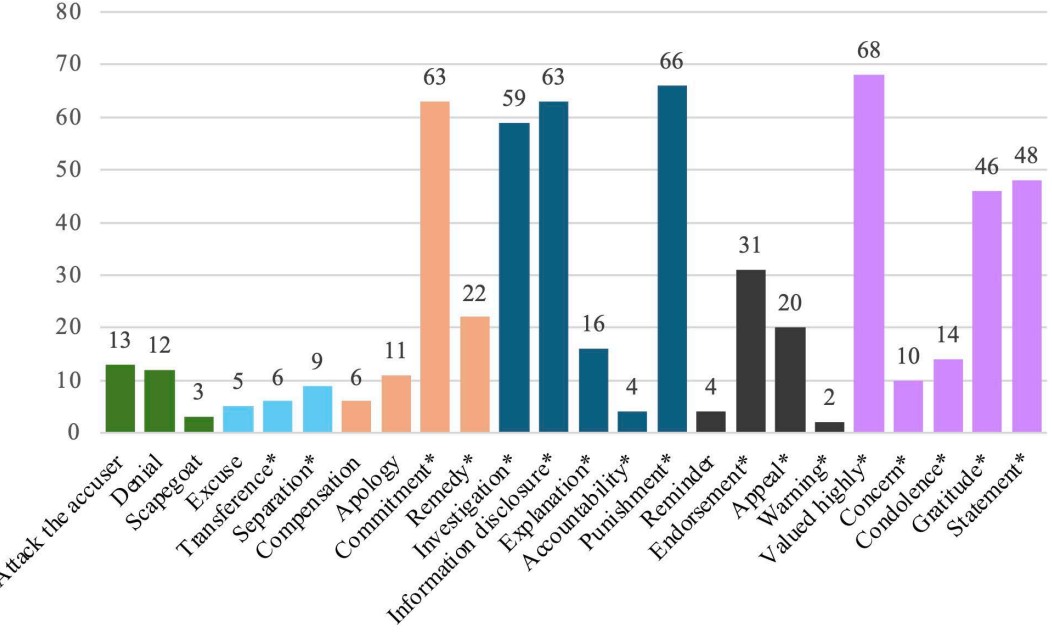

**Fig 1. Number of response strategies uses.**

the least are warning, scapegoat, accountability and reminder. Justification, ingratiation and victimage are not found to be used in the crisis response in Chinese universities.

### 4.3. Crisis clusters in Chinese universities

SCCT identified three crisis clusters according to the attribution of crisis responsibility:(1) the victim cluster has very weak attributions of crisis responsibility (natural disasters, workplace violence, product tampering and rumor) and the organization is viewed as a victim of the event with mild reputational threat; (2) the accidental cluster has minimal attributions of crisis responsibility (technical-error accident, technical-error product harm and challenge) and the event is considered unintentional or uncontrollable by the organization with moderate reputational threat and (3) the intentional cluster has very strong attributions of crisis responsibility (human-error accident, human-error product harm and organizational misdeed) and the event is considered purposeful with severe reputational threat [54].

The statistical results show that 82 out of 147 crisis response cases in Chinese universities in this study belong to the victim cluster, accounting for 55.8%, indicating that Chinese universities usually respond to crises with weak attributions of responsibility. The crisis of the victim cluster of Chinese universities has very weak attributions of crisis responsibility and mild reputational threat. In the victim cluster, the university is also a victim of the crisis. These crises include the following situation: rumor incidents, natural disasters and unexpected accidents which damaged the reputation of universities. Such as improper relations between professors and students, publishing inappropriate information on the Internet like false information, pornography, or speeches of disrupting public order by members of universities. For example, rumor appeared on the internet about multiple male students at Guangdong Vocational and Technical University of Business and Technology sexually assaulting a female student, and a law school professor at Sun Yat-Sen University had a messy personal life and being exposed online, causing damage to the reputation of the university.

34 cases belong to the accident cluster, accounting for 23.1%. The crisis of the accident cluster of Chinese universities has minimal attributions of crisis responsibility and moderate reputational threat. In the accidental cluster, the university actions leading to the crisis are unintentional without causing substantial harm or the university is blamed for the leadership responsibility of individual behavior that have a huge impact. For example, the incident of student abusing cats at Shandong University, although it was an individual act, triggered a wide discussion on the internet in China, Shandong University should bear the responsibility of leadership. Another representative incident is that Changsha University of Science and Technology library clean up students' examination materials arbitrarily, which belong to the improper behavior of the university, but did not cause personal injury.

31 cases belong to the intentional cluster, accounting for 21.1%. The crisis of the intentional cluster of Chinese universities has very strong attributions of crisis responsibility and severe reputational threat. In the intentional cluster, the university knowingly placed people at risk, took inappropriate actions or violated a law or regulation. These crises include the following situation: improper management of laboratory or security resulting in death or injury, the universities violate the rules of academic fairness and examination fairness which makes a huge impact and even causes damage to the credibility of the entire education system. Such as the canteen of Tongji University were suspected to purchase substandard pork, which threatened the lives of many students due to improper management of the university. There is also a case of Chen, an undergraduate student at Southwest Jiao Tong University, who cheating in the graduate exam, which seriously affected the fairness of the examination and enrollment procedures. Universities bear great responsibility for such incidents.

## 4.4. Applicable response strategies for different crisis clusters

SCCT emphasizes the adaptation of crisis response strategies to different situations. This study conducts a comparative analysis of the use of crisis response strategies in different situations. As Fig 2 shows, in the intentional cluster, the primary strategy of rebuild, including compensation, apology, commitment, and remedy, are most frequently employed. Chinese universities tend to use accommodative strategies when responsibility attribution is strong. Accountability and punishment strategies are also common, reflecting a similar accommodative approach. Endorsement, concern, and condolence are the most used secondary strategies.

For the accident cluster, the strategy of diminish, including excuse, transference, and separation are prevalent. These strategies are optimal in situations with moderate responsibility attribution. Additionally, explanation is a key primary strategy, and reminder and warning are common secondary strategies which are well-suited to crises in the accident cluster.

In the victim cluster, the primary strategy of deny, which includes attack the accuser, denial, and scapegoat, are widely used. Although the public might not hold the universities directly responsible for the crisis in this situation, restoring public trust depends on the universities' response. Deny is effective in stabilizing public emotions with minimal cost in situations with weak responsibility attribution. Appeal, valued highly, gratitude, and statement are secondary strategies frequently employed. Notably, investigation and information disclosure strategies were applicable across all crisis situations.

Therefore, the results of this study prove that the actual situation of crisis response in Chinese universities is consistent with the concept of SCCT, that is, the degree of accommodation

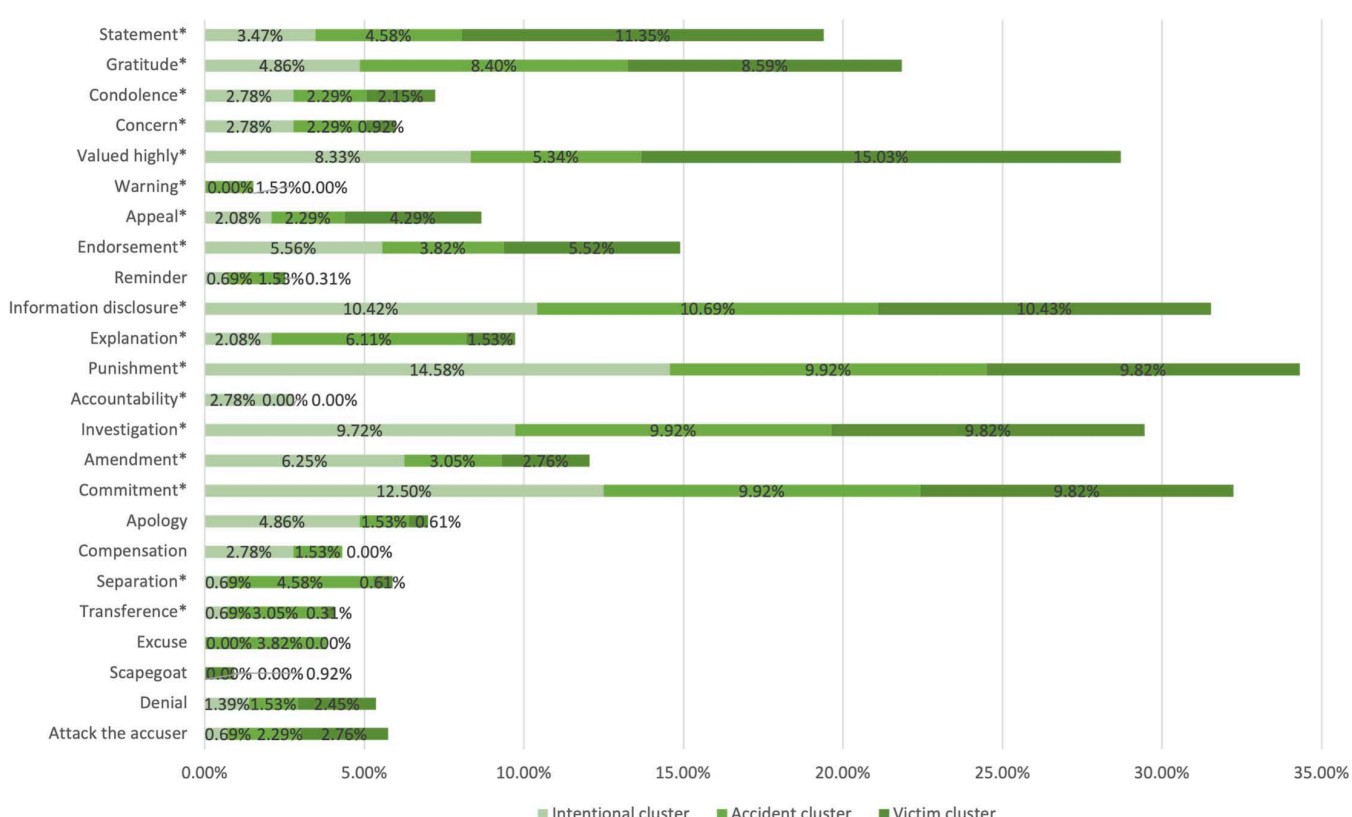

**Fig 2. The use of crisis response strategies in different clusters.**

of applicable strategies increases with the degree of responsibility. The conclusion of "the rebuild strategies are the safest crisis response for the reason of rebuild strategies addressing victims well" is not universally applicable [55], because using overly accommodating strategies in the inappropriate situations makes the cost higher to the organization [56–58] but may not see any increased yield in reputation protection [29]. Therefore, the effectiveness of the response to the crisis is not due to the unlimited accommodative response strategy, but to the adaptation of the response strategy to the category of the crisis. The results of this study provide a summary of recommendations for the use of crisis response strategies in Chinese universities.

## 5. Conclusion

The purpose of this study is to explore the crisis response strategies used by Chinese universities in external communication, and to establish a crisis response framework for Chinese universities in different situations by examining the differences in the use of crisis response strategies in different crisis clusters. To achieve this aim, this study adopts the method of qualitative thematic analysis and analyzes 182 response documents officially released by 125 Chinese universities in the face of 147 crises during the seven years from August 2016 to August 2023 to determine the crisis response strategies used by Chinese universities. This study further categorizes these 147 crises based on the crisis classification framework proposed by SCCT through content analysis, explores the differences in the crisis strategies used by Chinese universities in the face of different crises, and identifies the applicability of different strategies to different crisis clusters.

The results show that most of the crisis response strategies proposed by SCCT remain applicable to Chinese universities, such as attack the accuser, denial, scapegoat, excuse, compensation, apology and reminder. To some extent, this result shows the basic role and wide application of SCCT in the field of crisis response. However, this study found that Chinese universities have adopted many new strategies in crisis response. The strategies of transference [34], separation [33], remedy [31], information disclosure [35], explanation [35], accountability [35], punishment [35], endorsement [33] and gratitude [35] were found in the previous research from other researchers and also found to be used by Chinese universities in crisis response in this study. The strategies of commitment, investigation, appeal, warning, valued highly, concern, condolence and statement are new strategies which are discovered, defined and named in this study. A great deal of data of these strategies can be found in the response documents of Chinese universities. These outcomes indicate that the existing strategies are insufficient in response to all crisis situations, and the same crisis response strategy framework cannot be applied to all crises. It is vital for researchers to constantly explore and develop in theory under the background of different national conditions, contexts and crisis types.

This study further reveals that Chinese universities use different response strategies for different crises. The higher the degree of responsibility for the crisis, the more accommodative the crisis strategies applied by Chinese universities, which is consistent with the conclusion of SCCT. Using overly accommodating strategies in unnecessary situations will not only fail to achieve effective results, but will also increase the cost of the organization, damage its reputation, and arouse suspicion among stakeholders. For example, the strategies of apology, accountability or punishment used by Chinese universities in a crisis with little responsibility will not appease the public opinion, but rather arouse the public's suspicion about whether the university tries to use accommodative strategies to hide the information that has not been disclosed on the contrary. It shows that it is of great significance to identify the applicable strategies for different crises.

This study not only deepens the understanding of crisis response strategies in Chinese universities but also makes significant contributions to the application and development of SCCT globally. It will be elaborated in detail from theoretical implications and practical implications as follows.

## 5.1.  Theoretical implications

Firstly, this study enriches and expands the theoretical framework of SCCT. Through empirical research in the specific context of Chinese universities, it not only validates the basic framework of SCCT, but also further refines the classification of crises and the strategies that should be adopted for different crisis types. SCCT is recognized as a mature theoretical system in the field of crisis response, but in the nearly 20 years since it was proposed, many scholars have expanded the theory. As Coombs the founder of the theory notes, "an omnipotent list of crisis strategies are nonexistent". Combining the results of SCCT and other studies, this study found new strategies in crisis response in Chinese universities, named and defined these strategies in detail, further enriched the crisis response strategies, and injected new impetus for the development of the theory to a certain extent.

Secondly, this study exploratory explores the strategies used by Chinese universities in crisis response by focusing on the official response documents in all effective cases between the time when the official documents released by the Chinese Government requiring all kinds of institutions to respond to crises in August 2016 and the research carried out in August 2023, which provides a great deal of source information regarding crisis response in Chinese universities, showing how Chinese universities publish the response documents for crises on social media.

Finally, this study facilitates a more precise application and adaptation of SCCT in different cultural and contexts worldwide. As China's influence on the global stage continues to grow, research findings on crisis response in Chinese universities will attract more international attention. Conducting research on crisis response by applying SCCT within the Chinese context not only helps enhance the international applicability of SCCT but may also inspire more countries and regions to engage in research and practice related to this theory, thereby promoting its widespread application and recognition globally.

## 5.2.  Practical implications

This study has useful implications for the practitioners and policymakers of crisis response in Chinese universities. The results of this study not only provide strategy options for the practitioners and policymakers of crisis response in Chinese universities, but also, more importantly, as demanded by SCCT, established a link between crisis situations and crisis response strategies. Firstly, this study classifies the crises in Chinese universities based on the definition of crisis cluster classification proposed by SCCT, and basically formed a conceptual system of crisis classification of Chinese universities. The conceptual system of crisis classification may provide clues for Chinese universities to classify the crisis clusters in the future. However, the classification of crisis clusters and attribution of responsibility is challenging because it requires the inside knowledge and extensive experience of people who understand Chinese universities crisis and crisis responses. The coding process of this study is carried out by two coders independently, a doctoral student of public opinion crisis and a staff engaged in the work related to public opinion crisis response. The classification method improves the scientificity and reliability of the subsequent analysis results to a certain extent, and also provides a reference for future research.

Secondly, based on the classification of crisis clusters, this study analyzes the adaptability of crisis response strategies and situations in Chinese universities. A perfect list of crisis response

strategies, which is applicable to all kinds of crisis, is considered to be nonexistent, but what can be created is a list of useful crisis response strategies [55]. This study provides a useful framework to help Chinese universities choose the most appropriate response strategy, which are useful to both crisis managers and researchers. For the crisis managers in Chinese universities, it is vital to make informed choices about crisis response strategies based upon theoretically derived and empirically tested evidence rather than rely on hunches or recommendations for simple case studies[59]. In this respect, this research is crucial for the field of university crisis response, because different kinds of crises are happening frequently in universities all around the world and will continue to have an impact on the image of universities. In the age of social media development, it is of great significance for universities to carry out effective response in the face of crises.

The research findings of this study also contribute to other organizations or academic research worldwide. This study enhances the practicality and operability of SCCT, and its findings contribute to enabling universities, governments, and enterprises globally to develop and implement communication strategies more rapidly and effectively when facing crises. Furthermore, the research results of this study offer a foundation for cross-cultural comparisons. The research on crisis response strategies in Chinese universities provides a valuable case for cross-cultural comparisons within the context of SCCT. By comparing crisis management practices in Chinese universities with those in Western or other regional universities, it can reveal the similarities and differences in crisis communication strategies across different cultural and social backgrounds, thereby promoting the exchange and integration of crisis management knowledge globally.

## 6. Limitations and future directions

This study overlooks the impact of crisis history and prior relationship reputation on reputational threat. SCCT proposes that crisis history and prior relational reputation have both direct and indirect effect on the reputational threat posed by the crisis [55]. However, in the present study, crisis history and prior relational reputation are not considered. Given the study's large sample size, it resulted in a significant workload. Additionally, further consideration is needed regarding whether to treat each university as an independent unit, with different crises within the same university serving as variables, or to regard all universities as an alliance, utilizing the same type of crises across different universities as variables. Although this study has not explored this issue in detail, it is valuable to further investigate in future research.

Future research should explore the combinations of strategies employed in different crises. Across 147 crises in Chinese universities, a total of 601 response strategies were utilized, averaging 4.01 response strategies for each crisis, indicating that Chinese universities typically use a combination of strategies in crisis response. Are there any fixed combinations of these strategies? Are there any different patterns of strategies combination between different clusters? Further investigation of the combination of crisis response strategies in Chinese universities may promote SCCT and provide more detailed guidelines for crisis response in Chinese universities.

In the future, the attributions of crisis responsibility can be determined by news media and public opinion. The attribution of crisis responsibility defined by two coders with specialized knowledge through coding in this study still poses a certain degree of risk. SCCT believes that the public's attribution of responsibility for the crisis is largely influenced by the news media that how the news media frame (define) the crisis is an important consideration [55]. The attribution of responsibility for the crisis should examine the news media or conduct quantitative research on public opinion to increase the credibility of the findings in the future.

Furthermore, the strategies identified in this study initially demonstrate the strategies applied by universities as response entities in their crisis responses. However, the essence of crisis communication goes beyond the unilateral strategic choices of information issuers. Crisis response is a two-way process of perception management, which hinges on whether stakeholders (students, faculty, the public, etc.) truly receive, understand, and accept these strategies. In the future, the stakeholders' perceptions will be further taken into consideration, the findings of this research can serve as a baseline framework, against which future related studies can measure the degree of alignment or inconsistency in cognitive outcomes.

Overall, despite certain limitations, this study has extended SCCT both theoretically and practically. It enriched related research on SCCT and provides other researchers with empirical evidence on the crisis response of Chinese universities.

## Supporting information

**S1 Table. Example Sentences for Crisis Response Strategies.**
(DOCX)

## Acknowledgments

Thanks to the National University Online Public Opinion Monitoring Center for the help and support in obtaining the data set and data processing.

## Author contributions

**Conceptualization:** Liu Yongshi.

**Data curation:** Liu Yongshi, Hongtao Duan.

**Formal analysis:** Liu Yongshi, Hongtao Duan.

**Investigation:** Liu Yongshi.

**Methodology:** Liu Yongshi, Hongtao Duan.

**Supervision:** Deyi Gao.

**Visualization:** Liu Yongshi.

**Writing – original draft:** Liu Yongshi.

**Writing – review & editing:** Deyi Gao.

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
