## [Decision Letter · Decision Letter 0]

8 Aug 2024

PONE-D-24-14141Choosing the right strategies:An analysis of crisis response strategies in Chinese universitiesPLOS ONE

Dear Dr. Yongshi,

Thank you for submitting your manuscript to PLOS ONE. After careful consideration, we feel that it has merit but does not fully meet PLOS ONE’s publication criteria as it currently stands. Therefore, we invite you to submit a revised version of the manuscript that addresses the points raised during the review process.

We look forward to receiving your revised manuscript.

Kind regards,

Ahmed Meri, Ph.D.

Academic Editor

PLOS ONE

Journal Requirements:

2. In your Methods section, please include additional information about your dataset and ensure that you have included a statement specifying whether the collection and analysis method complied with the terms and conditions for the source of the data.

3. In the online submission form, you indicated that "Due to the number of real names involved in the data and the confidentiality agreement with the National University Online Public Opinion Monitoring Center, we are unable to release all the data. If editors and reviewers have questions about specific data, we can provide more data and do our best to provide more detailed explanations and instructions. The dataset is available from the corresponding author on reasonable request."

4. We notice that your supplementary tables are included in the manuscript file. Please remove them and upload them with the file type 'Supporting Information'. Please ensure that each Supporting Information file has a legend listed in the manuscript after the references list.

Reviewers' comments:

Reviewer's Responses to Questions

**Comments to the Author**

1. Is the manuscript technically sound, and do the data support the conclusions?

Reviewer #1: Partly

Reviewer #2: Yes

2. Has the statistical analysis been performed appropriately and rigorously? 

Reviewer #1: Yes

Reviewer #2: Yes

3. Have the authors made all data underlying the findings in their manuscript fully available?

Reviewer #1: Yes

Reviewer #2: No

4. Is the manuscript presented in an intelligible fashion and written in standard English?

Reviewer #1: Yes

Reviewer #2: Yes

5. Review Comments to the Author

Reviewer #1: Abstract/ it is important to clarify the methodology of the research.

Introduction/ the authors must indicate more details about the crises types in China and also which crises are the focus of the current study.

Gap and what makes this study different are not obvious in the introduction.

Literature review/ all universities mentioned in this part should by clarified by adding the country.

Methodology/ why the authors select 2016.08 to 2023.08? justification needs to be clarified.

More information about How the authors analyzed data are required.

Also, the two questions mentioned in the method section should be showed.

More details about how the authors conduct the content analysis are essential. The following reference could be helpful;

Salem, I. E., Elkhwesky, Z., & Ramkissoon, H. (2022). A content analysis for government’s and hotels’ response to COVID-19 pandemic in Egypt. Tourism and Hospitality Research, 22(1), 42-59.

Findings and discussion/

Tables should be less and summarized in an easy way.

Conclusion/ this part is very poor. So, it is important to divide it into two subtitles; theoretical contribution and practical implications.

The English language should be revised by a native speaker.

Reviewer #2: More LR needs to be added.

Details of the methodology need to be provided.

The basis of selection and choices in crises needs to be elaborated.

How study can be fruitful to policymakers needs to be added.

Check the referencing style.

6. PLOS authors have the option to publish the peer review history of their article (what does this mean? ). If published, this will include your full peer review and any attached files.

**Do you want your identity to be public for this peer review?** For information about this choice, including consent withdrawal, please see our Privacy Policy .

Reviewer #1: **Yes: ** Dr. Zakaria Elkhwesky

Reviewer #2: No

---

## [Author Response · Author response to Decision Letter 1]

27 Sep 2024

Dear Editor and Reviewers:

We sincerely thank you for your time and effort in reviewing this manuscript. Your professional comments have greatly helped to improve this study. We have followed your suggestions and conscientiously revised and responded to each comment. Thank you again for your valuable suggestions, and we sincerely hope that the revision of this manuscript can get your approval.

Regarding revisions required by the editor:

Thank you for the reminder. We checked the manuscript style, and we believe the style has already met PLOS ONE's style requirements.

2. In your Methods section, please include additional information about your dataset and ensure that you have included a statement specifying whether the collection and analysis method complied with the terms and conditions for the source of the data.

We declare that the collection and analysis method complied with the terms and conditions for the source of the data.

After anonymizing the names of all the people involved in the crisis, we agreed that the data would be available. We will do our best to meet the requirements of the journal. We have already uploaded all the data as supplementary information.

4. We notice that your supplementary tables are included in the manuscript file. Please remove them and upload them with the file type 'Supporting Information'. Please ensure that each Supporting Information file has a legend listed in the manuscript after the references list.

Thanks for the reminder. We are Sorry for our oversight. We will make corrections when the revised manuscript is submitted

Regarding revisions required by the reviewer#1:

1.Abstract/ it is important to clarify the methodology of the research.

Thanks to Dr. Zakaria Elkhwesky for your professional comments. According to your suggestion, the methodology of the research have been added in the abstract.

2.Introduction/ the authors must indicate more details about the crisis types in China and also which crises are the focus of the current study. Gap and what makes this study different are not obvious in the introduction.

Thanks for your professional comments. Details about the crisis types in China and crises which are the focus of the current study, also the distinction of this study have been added to the manuscript. Due to the large amount of content, it is not shown in this letter. Please review the revised manuscript. We hoped that the revised manuscript has been improved.

3.Literature review/ all universities mentioned in this part should by clarified by adding the country.

Thanks for the reminder. The universities mentioned in the prat of literature review have been clarified by add the country.

4.Methodology/ why the authors select 2016.08 to 2023.08? justification needs to be clarified.

Regarding the reason why we select 2016.08 to 2023.08 as the time range for case selection, we have added further explanations to the manuscript.

“In this study, representative crises on Chinese social media from 2016.08 to 2023.08 are selected as the research objects, since the notice of “Further Improving Response Capabilities to Government Affairs Crises in the Work of Openness of Government Affairs” became a landmark in China's crisis response, which issued in August 2016 by Chinese government to require clearly that “governments at all levels and their departments should attach great importance to the response to public opinion crisis in government affairs and implement the responsibility of response". The release of the notice standardized the public opinion crisis response and promoted the public opinion crisis response to become the normalization of the government and various institutions. Since then, the response to the public opinion crisis in Chinese universities has also begun to increase gradually. Therefore, this study selected crises between the time when the notice was released to the end time as the research objects of this study.”

More information about how the authors analyzed data are required.

Thanks to Dr. Zakaria Elkhwesky for your professional comments. After consideration, we did find that the section on data analysis in the initial manuscript was short of description in detail. Therefore, we have made revision in the manuscript. We have redescribed the process of data analysis in the manuscript. The main additions are as follows, minor changes were also made to the rest, which are reflected in the manuscript.

“In this study, two researchers participated in coding work. One is a doctoral student in public opinion crisis, and the other is a staff engaged in the work related to public opinion crisis response. In the first step, the strategies proposed in SCCT and references mentioned above are sorted out as the original identified codes and themes. The Chinese university response documents are coded in sentences, and potential new strategies that do not fit the predetermined codes are recorded in new codes. Generating codes requires marking interesting features of the data in a systematic way. Then, all the codes are collected and sorted, the strategies that fit the definitions are classified into the predetermined themes, and any potential new themes that do not fit the predetermined themes are temporarily placed in the miscellaneous category which is added to account for any incongruous evidence. The researchers investigate whether the themes have "central organizing concepts”. Researchers generate new themes and then name and define the new themes based on the central concepts.

For more clarification, it is significant to verify the transferability, dependability and confirmability of data in qualitative thematic analysis. The concepts of transferability, dependability and confirmability were thoroughly examined and proved in this study. Transferability is determined by achieving reference adequacy by archiving the second half of the dataset until the first half is analyzed, and then comparing the data in the second half with the first half. Dependability is verified by peer debriefing. One researcher provides another with detailed notes showing how he came to his findings, and another researcher provides an external check to ensure that the observations and interpretations of the data are valid. Confirmability is achieved by researchers detailing their notes in a systematic way to illustrate the connections between their data and findings. Through the verification, the coding scheme is basically determined.

In the second step, according to the coding scheme, coders were asked to code all the response documents sentence by sentence, but the same strategy used repeatedly in the same crisis was still recorded as once in the final count. To further ensure the full understanding and consistency of the coding scheme between the two coders, 20% of the documents were conducted preliminary test on request. Before the comprehensive implementation of the coding, any inconsistencies in coding or new discovered response strategies are introduced to a third researcher for further discussion until a consensus was reached.”

Also, the two questions mentioned in the method section should be showed.

Thank you for your suggestion. The two questions mentioned in the method section have already been showed in the revised manuscript.

More details about how the authors conduct the content analysis are essential. The following reference could be helpful; Salem, I. E., Elkhwesky, Z., & Ramkissoon, H. (2022). A content analysis for government’s and hotels’ response to COVID-19 pandemic in Egypt. Tourism and Hospitality Research, 22(1), 42-59.

We read the article in detail. The presentation of the methodology in this article is outstanding. We have made a lot of modifications to the part of method in the manuscript after referring to this article. Please check whether there has been any improvement of the part of method in the revised manuscript. Thank you for your comments and your reference, which has been an important guide for us.

5.Findings and discussion/ Tables should be less and summarized in an easy way.

Thank you for your suggestion. In the manuscript, first, we deleted Table 3, the text in the manuscript is sufficient to describe the content in this table. Second, we removed Table 4 to present it by Fig 2. which is more aesthetically pleasing and minimalist. Also, we deleted Table 5 and summarized it in a simple way. Finally, we found that the contents of 4.4 and 4.5 can be combined, and we have made some adjustments to the structure of the conclusions of these two parts, hoping that the revised manuscript will be better.

6.Conclusion/ this part is very poor. So, it is important to divide it into two subtitles: theoretical contribution and practical implications.

Thank you for your valuable advice. We read the conclusion part of the reference “Salem, I. E., Elkhwesky, Z., & Ramkissoon, H. (2022). A content analysis for government’s and hotels’ response to COVID-19 pandemic in Egypt. Tourism and Hospitality Research, 22(1), 42-59”in detail and studied the conclusion structure of the reference. We have rewritten the conclusion by divide it into two subtitles: theoretical implications and practical implications. Due to the large amount of content, it is not shown in this letter. Please review the revised manuscript. We hoped that the revised manuscript has been improved.

7.The English language should be revised by a native speaker.

Thanks to Dr. Zakaria Elkhwesky for your advice. We will review the language of this article and seek help from our peers.

Regarding revisions required by the reviewer#2:

1. More LR needs to be added.

Thanks for your professional comments. The manuscript has been extensively revised and the number of references has increased from 39 to 58.

2.Details of the methodology need to be provided.

Thank you for your valuable advice. We did find that the section on data analysis in the initial manuscript was short of description in detail. Therefore, we have made revision in the manuscript. We redescribe the process in qualitative thematic analysis of RQ1. The description of the content analysis of RQ2 is also supplemented. Due to the large amount of content, it is not shown in this letter. Please review the revised manuscript. We hoped that the revised manuscript has been improved.

3.The basis of selection and choices in crises needs to be elaborated.

Thank you for your suggestion. In this study, the basis of selection and choices in crises are constituted by 3 reasons as we mentioned in the part of data collection of the manuscript. First, the crisis has been on the list of hot topics on Sina Weibo, which proves that the crisis is with media attention and public discussion. Second, the crisis must be in the case-base of the National University Online Public Opinion Monitoring Center, this case-base is a collection of all the crises with topic sensitivity and event impact. Third, the public discussion of the crises which we selected in this study decreased significantly and faded within a week and no secondary public opinion crisis occurred after the issuance of the response document. We considered the response documents of Chinese universities in these crises to be initially effective in this study. In future studies, we will further improve the research by empirically analyzing the effectiveness of responses through public reaction.

4.How study can be fruitful to policymakers needs to be added.

Thank you for your suggestion. Regarding how study can be fruitful to policymakers, we have rewritten the part of conclusion. Especially in the part of practical implication, we added some content, hoping to improve the manuscript.

“This study has useful implications for the practitioners and policymakers of crisis response in Chinese universities. The results of this study not only provide strategies options for the practitioners and policymakers of crisis response in Chinese universities, but more importantly, according to the demand of SCCT, a link between crisis situations and crisis response strategies should be established.”

5.Check the referencing style.

Thanks for reminding. We have re-checked the referencing style. We are Sorry for our oversight.

Thanks again to all the editors and reviewers for your professional suggestions on the revision of this manuscript. We hope that our revision of the manuscript can get your approval. If there are any suggestions still worth making, we would appreciate your reminder and your patience. We'll do everything we can to make the manuscript the best it can be.

---

## [Decision Letter · Decision Letter 1]

19 Nov 2024

PONE-D-24-14141R1Choosing the right strategies:An analysis of crisis response strategies in Chinese universitiesPLOS ONE

Dear Dr. Yongshi,

Thank you for submitting your manuscript to PLOS ONE. After careful consideration, we feel that it has merit but does not fully meet PLOS ONE’s publication criteria as it currently stands. Therefore, we invite you to submit a revised version of the manuscript that addresses the points raised during the review process.

We look forward to receiving your revised manuscript.

Kind regards,

Ahmed Meri, Ph.D.

Academic Editor

PLOS ONE

Journal Requirements:

Reviewers' comments:

Reviewer's Responses to Questions

**Comments to the Author**

1. If the authors have adequately addressed your comments raised in a previous round of review and you feel that this manuscript is now acceptable for publication, you may indicate that here to bypass the “Comments to the Author” section, enter your conflict of interest statement in the “Confidential to Editor” section, and submit your "Accept" recommendation.

Reviewer #2: All comments have been addressed

2. Is the manuscript technically sound, and do the data support the conclusions?

Reviewer #2: Yes

3. Has the statistical analysis been performed appropriately and rigorously? 

Reviewer #2: Yes

4. Have the authors made all data underlying the findings in their manuscript fully available?

Reviewer #2: Yes

5. Is the manuscript presented in an intelligible fashion and written in standard English?

Reviewer #2: No

6. Review Comments to the Author

Reviewer #2: The revisions should be highlighted and put as the main text, not after the original manuscript. In text, citations should be given rather than the numbers.

The revised introductions need to be rephrased. It should be in the form of academic writing, not as a normal conversation.

Elaborate more on the technique used for the content analysis.

Refer to other seminal work on content and thematic analysis.

e.g.,

Liñán, F., & Fayolle, A. (2015). A systematic literature review on entrepreneurial intentions: citation, thematic analyses, and research agenda. International entrepreneurship and management journal, 11, 907-933.

Kim, H., & So, K. K. F. (2022). Two decades of customer experience research in hospitality and tourism: A bibliometric analysis and thematic content analysis. International Journal of Hospitality Management, 100, 103082.

Vaismoradi, M., Turunen, H., & Bondas, T. (2013). Content analysis and thematic analysis: Implications for conducting a qualitative descriptive study. Nursing & health sciences, 15(3), 398-405.

Thompson, J. (2022). A guide to abductive thematic analysis.

Recheck referencing style. It is better if APA style is used as per the guidelines.

7. PLOS authors have the option to publish the peer review history of their article (what does this mean? ). If published, this will include your full peer review and any attached files.

**Do you want your identity to be public for this peer review?** For information about this choice, including consent withdrawal, please see our Privacy Policy .

Reviewer #2: No

---

## [Author Response · Author response to Decision Letter 2]

6 Dec 2024

Dear Editor and Reviewer:

We sincerely thank you for the time and effort you spent providing professional comments on our articles. As you are concerned, there are still several issues that need to be addressed. We have carefully reconsidered these problems and revised them one by one according to your suggestions. We sincerely hope that the revision of this manuscript can get your approval.

Regarding revisions required by the editor:

1.Please review your reference list to ensure that it is complete and correct. If you have cited papers that have been retracted, please include the rationale for doing so in the manuscript text, or remove these references and replace them with relevant current references. Any changes to the reference list should be mentioned in the rebuttal letter that accompanies your revised manuscript. If you need to cite a retracted article, indicate the article’s retracted status in the References list and also include a citation and full reference for the retraction notice.

Thanks to the editor for the reminder. We re-examined the references individually to confirm that they had not been retracted. In addition, we have added several references during the revision of the manuscript. The added references are listed below：

[1] Leibold J. Blogging alone: China, the internet, and the democratic illusion? The Journal of Asian Studies. 2011;70 (4):1023–1041.

[2] Weng SH. Participation-response model: An analytical framework of the government response to internet political participation for decision-making. Journal of Public Administration. 2014;7(05):109-130+191.

[3] Sun LM. The rise of civil rights consciousness: A largely internet-based observation. Journal of the Central Institute of Socialism. 2010;(03):99-103.

[55] Vaismoradi M, Turunen H, Bondas T. Content analysis and thematic analysis: Implications for conducting a qualitative descriptive study. Nursing and Health Science. 2013;15,398-405.

[56] Hayes AF, Krippendorff K. Answering the call for a standard reliability measure for coding data. Communication Methods and Measures. 2007; 1:77-89.

Also, we have double-checked the references format and carefully revised it to ensure that they have met the journal's requirements for the Vancouver style.

Regarding revisions required by the reviewer #2:

1.The revisions should be highlighted and put as the main text, not after the original manuscript. In text, citations should be given rather than the numbers.

Thank to the reviewer for the reminder. For the main parts of the revision, we have highlighted it in green font in the main text. We have carefully considered and revised the article according to your valuable suggestions. In addition, citations have been modified as required by the journal and your suggestion, also we highlighted the changes to citation in the article for your review. We are sorry for our carelessness and thank you again for your reminder.

2.The revised introductions need to be rephrased. It should be in the form of academic writing, not as a normal conversation.

Thanks for your professional comments. We have reconsidered and rewritten the part of introduction. The revised introduction is more academic than introductory. We deleted the simple introduction of the type of public opinion crisis on the internet in China, and strengthened the explanation of why the internet has become the most important channel for Chinese citizens to express their opinions and demands:

“Unlike some Western developed countries, they have many offline institutionalized channels for citizen participation such as public hearings and citizens’ assemblies in addition to online participation. In China, there is still a contradiction between the narrow channels for citizen participation and the enthusiasm of modern citizens for political participation. Therefore, the convenience of social media has made it the most convenient way for Chinese citizens to express opinions and seek solutions to problems from the government. The expression of opinions and demands through online participation has become the most active part of Chinese citizens in the public sphere of the internet.”

Your valuable suggestion helps to improve the quality of our article, and we hope the revision can get your approval.

3.Elaborate more on the technique used for the content analysis.

Thanks to reviewer for your professional advice. After re-examination, we did find that the description of the content analysis section was not detailed enough, and we omitted many details during the writing process, which were not shown in the article. We have re-described this part in detail, hoping to explain the process of content analysis clearly. The main modifications are as follows:

“Content analysis is one of the most important research techniques in social sciences to analyze data with a specific context. Thus, when using content analysis, the researchers should have a broader understanding of the context. Content analysis provides a systematic and objective means to make valid inferences from verbal, visual, or written data to describe and quantify specific phenomena in a conceptual form. There are three main steps in content analysis: (a)obtaining the sense of the whole data, selecting the unit of analysis, (b)creating categories and open coding, (c)reporting the analysis process and the results through conceptual system.

In this section, taking a case as a unit, the two coders coded independently to divide Chinese universities crises into three clusters after fully understanding the relevant concepts of crisis classification based on SCCT and the research requirements of classification category. The victim cluster, the accident cluster and the intentional cluster are the three first-level codes of this coding process. First, 20 cases were randomly selected for independent coding. After coding, consistency judgment is carried out. A reliability of 0.863 is achieved using Krippendorf’s alpha, which is higher than the standard of 0.7 and shows that two coders have a high degree of agreement in understanding data, context, and coding concepts in the content analysis. Also, the sample size of preliminary test meets the requirement of 10%-20% of the total samples. All remaining disagreements were completely discussed prior to the next implementation of coding until a consensus was reached. In the second round, 50 cases were randomly selected from the remaining cases for independent coding. The coding with inconsistent results were discussed prior to the next implementation of coding until a consensus was reached. In the third round, the remaining 77 cases were coded independently, and the coding with inconsistent results were discussed until a consensus was reached. The discussion after each round of coding makes the next round of coding less disagreement, and the conceptual system of crisis classification of Chinese universities are basically formed, which provides clues for Chinese universities to classify the crisis clusters quickly and reliably. After three rounds of coding, 147 cases were finally classified by the degree of responsibility.”

As Vaismoradi M, Turunen H and Bondas T (2013) proposed, “the creativity of the researcher for presenting the result in terms of conceptual system is encouraged in the content analysis”. Actually, in the coding process of content analysis, the researchers gradually formed a conceptual framework for crisis classification of Chinese universities, and finally divided 147 cases into three clusters according to this conceptual system. The conceptual system further refines the brief definition of crisis cluster classification of Chinese universities based on theoretical framework of SCCT. The specific content has been added to “4.3 Crisis cluster in Chinese universities”. The summary is as follows for the your review：

“According to the degree of responsibility attribution, the crises in Chinese universities can be divided into three clusters: (1) the victim cluster has very weak attributions of crisis responsibility and mild reputational threat. In the victim cluster, the university is also a victim of the crisis. These crises include the following situation: rumor incidents, natural disasters and unexpected accidents which damaged the reputation of universities; (2) the accidental cluster has minimal attributions of crisis responsibility and moderate reputational threat. In the accidental cluster, the university actions leading to the crisis are unintentional without causing substantial harm or the university is blamed for the leadership responsibility of individual behavior that have a huge impact. (3) the intentional cluster has very strong attributions of crisis responsibility and severe reputational threat. In the intentional cluster, the university knowingly placed people at risk, took inappropriate actions or violated a law or regulation. These crises include the following situation: improper management of laboratory or security resulting in death or injury, the universities violate the rules of academic fairness and examination fairness which makes a huge impact and even causes damage to the credibility of the entire education system.”

4.Refer to other seminal work on content and thematic analysis.

e.g.,

Liñán, F., & Fayolle, A. (2015). A systematic literature review on entrepreneurial intentions: citation, thematic analyses, and research agenda. International entrepreneurship and management journal, 11, 907-933.

Kim, H., & So, K. K. F. (2022). Two decades of customer experience research in hospitality and tourism: A bibliometric analysis and thematic content analysis. International Journal of Hospitality Management, 100, 103082.

Vaismoradi, M., Turunen, H., & Bondas, T. (2013). Content analysis and thematic analysis: Implications for conducting a qualitative descriptive study. Nursing & health sciences, 15(3), 398-405.

Thompson, J. (2022). A guide to abductive thematic analysis.

Thanks to reviewer for your professional support and guidance. We spent a lot of time reading these references recommended by reviewer in detail and got a deeper understanding of content analysis and thematic analysis. These articles provide a detailed introduction to content analysis and thematic analysis. Based on these seminal literatures on the two methods, first, we re-examined the thematic analysis process in our article to ensure its accuracy and standardization. Second, we have also re-examined the content analysis process in our article and supplementary descriptions of the omitted details are provided, as you suggested. The references you recommended will also help us maintain academic rigor and normativity in implementing content analysis and thematic analysis in our research in the future. Thank you again for your professional advice.

5.Recheck referencing style. It is better if APA style is used as per the guidelines.

Thank you for your suggestion. We reviewed again the reference requirements on the journal page. According to the journal requirements, the Vancouver style should be use: “PLOS uses the reference style outlined by the International Committee of Medical Journal Editors (ICMJE), also referred to as the ‘Vancouver’ style”. Therefore, we have not changed the style in this revision. However, in the process of re-checking, we confirmed that there were many formatting errors, and we have revised them one by one. Thanks again for the reviewer's reminder.

Finally, according to the journal requirement that the manuscript should be presented in an intelligible fashion and written in standard English, we have asked for help from professional language polishing service, hoping to improve the language of this article. The certificate is as follows:

Thanks again to editor and reviewer for your professional suggestions on the revision of this manuscript. Your valuable comments have greatly improved this manuscript. We have carefully revised the manuscript according to your suggestions and hope that our revision can get your approval. If there are any suggestions still worth making, we would appreciate your reminder and your patience. We promise that we will do everything we can to make the manuscript the best it can be.

---

## [Decision Letter · Decision Letter 2]

20 Dec 2024

PONE-D-24-14141R2Choosing the right strategies:An analysis of crisis response strategies in Chinese universitiesPLOS ONE

Dear Dr. Yongshi,

Thank you for submitting your manuscript to PLOS ONE. After careful consideration, we feel that it has merit but does not fully meet PLOS ONE’s publication criteria as it currently stands. Therefore, we invite you to submit a revised version of the manuscript that addresses the points raised during the review process.

We look forward to receiving your revised manuscript.

Kind regards,

Ahmed Meri, Ph.D.

Academic Editor

PLOS ONE

Journal Requirements:

Reviewers' comments:

Reviewer's Responses to Questions

**Comments to the Author**

1. If the authors have adequately addressed your comments raised in a previous round of review and you feel that this manuscript is now acceptable for publication, you may indicate that here to bypass the “Comments to the Author” section, enter your conflict of interest statement in the “Confidential to Editor” section, and submit your "Accept" recommendation.

Reviewer #2: All comments have been addressed

Reviewer #3: All comments have been addressed

2. Is the manuscript technically sound, and do the data support the conclusions?

Reviewer #2: Partly

Reviewer #3: Yes

3. Has the statistical analysis been performed appropriately and rigorously? 

Reviewer #2: Yes

Reviewer #3: Yes

4. Have the authors made all data underlying the findings in their manuscript fully available?

Reviewer #2: Yes

Reviewer #3: Yes

5. Is the manuscript presented in an intelligible fashion and written in standard English?

Reviewer #2: Yes

Reviewer #3: Yes

6. Review Comments to the Author

Reviewer #2: The abstract is informative but could better emphasize the study's practical implications and theoretical contributions. The identification of new strategies should be explicitly highlighted as an extension of SCCT.

The introduction successfully sets the context but includes some verbose sections.The discussion of the public’s enthusiasm for political participation can be streamlined to maintain focus on crisis response.

The gap in literature could be more concisely articulated to underline the necessity of this research.

The SCCT framework is adequately explained, but the integration of new strategies into the framework appears to be forced.

Coder bias, or cultural context influencing the interpretation of data, is not discussed adequately. The rationale for excluding third-party narratives is valid but should be elaborated with respect to the exclusion of stakeholder perceptions.

The findings are repetitive in parts. The description of strategies used in each crisis cluster could be condensed without losing essential details.

The discussion section does not sufficiently explore the implications of the findings for global applications of SCCT.

The manuscript has undergone professional editing; minor grammatical issues and awkward phrasing persist, such as "unnecessary situations" (page 23) and "radically increasing" (page 14). Sentence structures could be varied to improve readability.

The conclusion reiterates key findings. It should also emphasize their significance for theory and practice. Additionally, it should provide scope for future research.

Reviewer #3: Dear Authors,

Please find the following comments to be revised in your manuscript:

1. ROWS 23-25, needs more clarification by adding some values that support the topic. Also, you should highlight the most adopted strategies by universities (maximum no. of response).

2. ROWS 31-36, needs to be reformulated.

3. Table 1, case No. 9, Add an explanation to this case that shows its strength.

4. ROW 302, (shown in S1 Table) …It is preferable to refer to the section and page.

5. Table 5, rearrange the table.

6. ROW 399, numeric values should be contained.

7. PLOS authors have the option to publish the peer review history of their article (what does this mean? ). If published, this will include your full peer review and any attached files.

**Do you want your identity to be public for this peer review?** For information about this choice, including consent withdrawal, please see our Privacy Policy .

Reviewer #2: No

Reviewer #3: **Yes: ** Zahraa Talib

---

## [Author Response · Author response to Decision Letter 3]

23 Jan 2025

Dear Editor and Reviewers:

We are deeply grateful and sincerely appreciate your generous investment of valuable time and energy in providing professional and in-depth review comments on our article. Even after two rounds of meticulous revisions, you maintained a rigorous attitude, carefully reviewed the manuscript again, and put forward detailed and invaluable suggestions. We hold your professionalism in the highest esteem and have learned invaluable academic attitudes from you. Based on your expert insights, we have once again reviewed the manuscript and made targeted improvements to its shortcomings. We eagerly anticipate and sincerely hope that the revised manuscript will meet with your approval and endorsement.

Furthermore, as we usher in the new year, we would like to take this opportunity to extend our warmest New Year wishes to the editor and reviewers of this article. Wish you all the best in 2025!

Regarding revisions required by the editor:

1.Please review your reference list to ensure that it is complete and correct. If you have cited papers that have been retracted, please include the rationale for doing so in the manuscript text, or remove these references and replace them with relevant current references. Any changes to the reference list should be mentioned in the rebuttal letter that accompanies your revised manuscript. If you need to cite a retracted article, indicate the article’s retracted status in the References list and also include a citation and full reference for the retraction notice.

Thanks to the editor for the reminder. We re-examined the references individually to confirm that they had not been retracted. In addition, we have deleted several references during the revision of the manuscript. The deleted references are listed below:

[3] Leibold J. Blogging alone: China, the internet, and the democratic illusion? The Journal of Asian Studies. 2011;70 (4):1023–1041.

[4] Weng SH. Participation-response model: An analytical framework of the government response to internet political participation for decision-making. Journal of Public Administration. 2014;7(05):109-130+191.

[5] Sun LM. The rise of civil rights consciousness: A largely internet-based observation. Journal of the Central Institute of Socialism. 2010;(03):99-103.

Regarding revisions required by the reviewer #2:

1.The abstract is informative but could better emphasize the study's practical implications and theoretical contributions. The identification of new strategies should be explicitly highlighted as an extension of SCCT.

Firstly, we would like to express our sincere gratitude to the reviewer for acknowledging and recognizing our work. Secondly, based on your comments, we have revised the abstract to emphasize the theoretical contributions and practical significance of the research, as well as to highlight that the new strategy is an extension of SCCT. The main modifications of abstract are as follows:

“The findings revealed that Chinese universities have adopted 17 new strategies beyond the strategies proposed by SCCT, highlighting a discrepancy between the theoretical ground of the theory and the application of strategies in real-life crisis response of Chinese universities. Furthermore, the results revealed the applicable strategies for different crisis clusters, which contributed to construct the framework of crisis response strategies for Chinese universities in different situations. This study expanded SCCT theoretically by enriching the crisis response strategies while practically improving the applicability of SCCT in Chinese universities. The findings provided guiding significance to the crisis response for Chinese universities in different crisis situations, also its innovative strategies provided empirical evidence for other organizations in different cultural contexts in crisis response.”

2.The introduction successfully sets the context but includes some verbose sections. The discussion of the public’s enthusiasm for political participation can be streamlined to maintain focus on crisis response.

Thanks to reviewer for your acknowledgment and suggestions. We have removed the lengthy discussion on public enthusiasm for political participation and emphasized the focus on crisis response. Your professional advice has made the introduction more concise and clearer. The main modifications of introduction are as follows:

Firstly, we deleted the verbose sections of“the internet has become the most important channel for Chinese citizens to express their opinions and demands. Unlike some Western developed countries, they have many offline institutionalized channels for citizen participation such as public hearings and citizens’ assemblies in addition to online participation. In China, there is still a contradiction between the narrow channels for citizen participation and the enthusiasm of modern citizens for political participation. Therefore, the convenience of social media has made it the most convenient way for Chinese citizens to express opinions and seek solutions to problems from the government. The expression of opinions and demands through online participation has become the most active part of Chinese citizens in the public sphere of the internet”.

Secondly, we have redescribed the first half of the introduction. The revised introduction maintain focus on crisis response as you suggested:

“With the development of social media in China, the Chinese public has increasingly gained the right to publish information and opinions. Public's awareness of political participation has been continuously enhanced, and the active degree of public speech and enthusiasm for political participation have reached an unprecedented height. Public opinion crises on social media have frequently occurred in China. In recent years, numerous crises in the fields of public health, food safety, medical malpractice, government management, product quality, security incidents, corporate image have aroused widespread public concern and discussion. The occurrence of public opinion crisis makes the relevant responsible parties under immense pressure to crises response. When public opinion crises occur, if the government, institutions, and enterprises fail to respond reasonably, it will cause great harm to their own image, lead to public opinion crises, and reduce credibility. Obviously, public opinion crisis response has become an important research topic at present.

Academics have shown great interest in research of these public opinion crises in China, and the relevant researches mainly focus on three points. First, researchers deeply discussed the interaction between government crisis management and public opinion, and analyzed the influence of public opinion on government management strategies and how the government effectively responds to these public opinions. Second, some researchers are committed to revealing the evolution law of public opinion, providing valuable insights for formulating effective response strategies through big data and computer models. Third, many researchers have conducted research on how enterprises and brands respond to negative public opinions on Chinese social media during crises. The research on public opinion crisis and crisis response is very important, which not only has profound theoretical value, but also shows its indispensable importance in reality.”

3.The gap in literature could be more concisely articulated to underline the necessity of this research.

Thanks to reviewer for your professional advice. According to your advice, we redescribed the gap of literature and underlined the necessity of this research.

Firstly, we deleted the verbose sections of “There are quite few studies taking Chinese universities as the research object. The research on response strategies in China mainly focuses on government administrative departments, enterprises, and brands. Also, most of the research on crisis response in Chinese universities is from a macro perspective and there are short of empirical research.”

Secondly, it was concisely expressed as “However, despite the prevalence of crises in Chinese universities, there is limited research on how to respond. In particular, empirical studies on crisis response in Chinese universities, are almost nonexistent”.

Also, we further underlined the necessity of this research that “the findings of this study provide a comprehensive and definitive answer to the question that has yet to be addressed in current research: what strategies are used in the public opinion crisis response in Chinese universities and what strategies are applicable to different types of crises”.

4.The SCCT framework is adequately explained, but the integration of new strategies into the framework appears to be forced.

Regarding your concern that the integration of the new strategies into this framework appears to be forced, we understand your perspective. Please allow us to explain in detail. In fact, in many studies on crisis response strategies based on the Situational Crisis Communication Theory (SCCT), researchers often incorporate new strategies into the SCCT framework. Take several articles mentioned in the literature review section of this study as examples. In fact, the integration of new strategies into the SCCT framework in this study is based on the following research findings.

Castonguay and Lowes (2022) [39] conducted a thematic analysis of the National Football League's (NFL) response to the concussion crisis and found that the NFL employed some new strategies, among which they integrated the strategy of “organizational change” into the “rebuild” strategy within the SCCT framework.

Kim and Liu (2012) [37] applied SCCT to investigate how 13 corporate and government organizations responded to the first phase of the 2009 flu pandemic through a quantitative content analysis, and expanded SCCT's response strategy options of “enhancing” and “transferring” and integrates them to a new theme called "other strategy”.

Lai and Tang (2018) [38] discovered six new strategies not mentioned by SCCT by analyzing 64 cases of Chinese public opinion crisis. They integrated the new strategies of “gratitude”, “accountability”, “punishment”, and “explanation” into the “primary strategies” and incorporated the new strategies of “information disclosure” and “attitude indication” into the “secondary strategies” of the SCCT framework.

Liu (2010) [35] applied SCCT to identify how organizations and individuals can effectively respond to racially charged crises through a content analysis of 104 response documents and 144 newspaper articles. He expanded SCCT by identifying several strategies that organizations and individuals used to respond to racially charged crises that were currently not included in the theory. In his study, the new strategy of “ignore” was integrated into the “deny” strategy within the SCCT framework, the new strategy of “separation” was integrated into the “diminish” strategy, and the new strategy of “transcendence” was integrated into the “rebuild” strategy. Additionally, he introduced a new theme called “reinforce” and incorporated a new strategy called “endorsement” into it (as shown in the figure below).

5.Coder bias, or cultural context influencing the interpretation of data, is not discussed adequately. The rationale for excluding third-party narratives is valid but should be elaborated with respect to the exclusion of stakeholder perceptions.

Thank you for your valuable suggestion. As pointed out by reviewer, coder bias, which refers to the influence of cultural context on data interpretation, does indeed exist. While this manuscript mentions it, the discussion is not sufficient. Therefore, we have made some revisions to address this issue:

“due to the influence of cultural context on the interpretation of data, it is worth noting that although the explanations and the example sentence for each strategy in this study were translated into English, researchers actually used Chinese during the analysis, because analysis in the original language of the document can fully reveal the use of strategies in the particular context, so as to ensure the accuracy and effectiveness of the analysis results.”

Besides, regarding your comment, "the rationale for excluding third-party narratives is valid but should be elaborated with respect to the exclusion of stakeholder perceptions," we have made the following modifications accordingly.

“As the purpose of this research is to determine how the Chinese universities communicated about crises in their external communications, news articles or interviews of school staff were excluded to avoid the subjectivity of a third party in the reporting and framing of the universities’ responses. All documentary evidence is officially released by universities. Meanwhile, due to stakeholders perceptions may be influenced by their own interests, and potential emotional factors and biases towards the organization, in order to focus on the response strategies of Chinese universities and maintain the objectivity of the research, the influence of stakeholders perceptions are also excluded in this study.”

6.The findings are repetitive in parts. The description of strategies used in each crisis cluster could be condensed without losing essential details.

Thank you for your suggestion. Upon your reminder, we have reviewed the findings section of the article again and found that there are indeed repetitive parts, especially in section 4.4. Therefore, we have carefully revised this section, and the specific revisions are as follows:

Firstly, we deleted the redundant parts: “Based on the research above, this study provides a summary of recommendations for the use of crisis response strategies in Chinese universities. For the crises of victim cluster, it is applicable to use the strategies of attack the accuser, denial, scapegoat, endorsement, appeal, valued highly gratitude and statement. For the crises of accident cluster, the strategies of excuse, transference, separation, explanation, reminder, warning, concern, condolence and gratitude are the reasonable choice in response. For the crises of intentional cluster, the strategies of compensation, apology, commitment, remedy, accountability, punishment, endorsement, concern and condolence are suitable in this situation. The study also reveals that the strategies of investigation and information disclosure can be applied to any situation in the public opinion crisis response in Chinese universities.”

Secondly, we have condensed the description of the strategies used in each crisis cluster without losing essential details. We compress the length by using more concise expressions like “Therefore, it is applicable to use these strategies for the crises of Chinese universities in the victim cluster” while preserving the original meaning. Due to the extensive length of the modifications, specific changes have been marked in different colored fonts in the manuscript. We kindly request the reviewer to review the manuscript and hope that the revised findings will meet your satisfaction.

7.The discussion section does not sufficiently explore the implications of the findings for global applications of SCCT.

Thank you to the reviewer for your professional and meticulous comments. Based on your suggestions, we have supplemented information regarding the impact of our research findings on the global application of SCCT. Due to the excessive length, only the added contents are shown in this response letter. The specific additions are as follows:

“This study not only deepens the understanding of crisis response strategies in Chinese universities but also makes significant contributions to the application and development of SCCT globally. This study enriches and expands the theoretical framework of SCCT. Through empirical research in the specific context of Chinese universities, it not only validates the basic framework of SCCT, comprehensively explores the strategies used by Chinese universities in crisis response but also further refines the classification of crises and the strategies that should be adopted for different crisis types.

This study facilitates a more precise application a

---

## [Decision Letter · Decision Letter 3]

25 Feb 2025

PONE-D-24-14141R3Choosing the right strategies:An analysis of crisis response strategies in Chinese universitiesPLOS ONE

Dear Dr. Yongshi,

Thank you for submitting your manuscript to PLOS ONE. After careful consideration, we feel that it has merit but does not fully meet PLOS ONE’s publication criteria as it currently stands. Therefore, we invite you to submit a revised version of the manuscript that addresses the points raised during the review process.

We look forward to receiving your revised manuscript.

Kind regards,

Ahmed Meri, Ph.D.

Academic Editor

PLOS ONE

Journal Requirements:

Reviewers' comments:

Reviewer's Responses to Questions

**Comments to the Author**

1. If the authors have adequately addressed your comments raised in a previous round of review and you feel that this manuscript is now acceptable for publication, you may indicate that here to bypass the “Comments to the Author” section, enter your conflict of interest statement in the “Confidential to Editor” section, and submit your "Accept" recommendation.

Reviewer #2: All comments have been addressed

Reviewer #3: All comments have been addressed

2. Is the manuscript technically sound, and do the data support the conclusions?

Reviewer #2: Yes

Reviewer #3: Yes

3. Has the statistical analysis been performed appropriately and rigorously? 

Reviewer #2: Yes

Reviewer #3: Yes

4. Have the authors made all data underlying the findings in their manuscript fully available?

Reviewer #2: Yes

Reviewer #3: Yes

5. Is the manuscript presented in an intelligible fashion and written in standard English?

Reviewer #2: Yes

Reviewer #3: Yes

6. Review Comments to the Author

Reviewer #2: The abstract provides a good summary. State explicitly how university administrators can use these findings in real crisis situations. Provide a sentence on how universities can incorporate these new strategies into their crisis response plans.

The paper justifies excluding third-party narratives, but it does not fully address why stakeholder perceptions were not considered. Since crisis communication is about perception management, future research should examine how students, faculty, and the public perceive these responses.

The newly identified strategies are mapped onto existing SCCT categories, but some seem to be stand-alone approaches rather than fitting neatly into SCCT. Instead of force-fitting into SCCT, consider discussing how these strategies might complement or challenge the existing framework.

Condense the discussion of response strategies per crisis cluster to avoid redundancy.

Table 5 requires restructuring for clarity. Consider sentence variation to improve readability.

Reviewer #3: No further comments

7. PLOS authors have the option to publish the peer review history of their article (what does this mean? ). If published, this will include your full peer review and any attached files.

**Do you want your identity to be public for this peer review?** For information about this choice, including consent withdrawal, please see our Privacy Policy .

Reviewer #2: No

Reviewer #3: **Yes**

---

## [Author Response · Author response to Decision Letter 4]

27 Feb 2025

Dear Editor and Reviewers:

We are immensely grateful for the precious time and effort you have dedicated to providing professional and in-depth comments on our article. Based on your invaluable suggestions, we have once again carefully reviewed and revised the article. We believe that after 4 meticulous rounds of revisions, the article has significantly improved. We sincerely hope to gain your recognition and obtain the opportunity for publication.

Thank you once again for the dedication of the editors and reviewers and wish you all the best!

Regarding revisions required by the editor:

1.Please review your reference list to ensure that it is complete and correct. If you have cited papers that have been retracted, please include the rationale for doing so in the manuscript text, or remove these references and replace them with relevant current references. Any changes to the reference list should be mentioned in the rebuttal letter that accompanies your revised manuscript. If you need to cite a retracted article, indicate the article’s retracted status in the References list and also include a citation and full reference for the retraction notice.

Thanks to the editor for the reminder. We re-examined the references individually to confirm that they had not been retracted. Furthermore, as this is a minor revision with limited changes, only one reference was removed during the revision process:

[59]Siomkos GJ, Kurzbard G. The hidden crisis in product harm crisis management. European Journal of Marketing. 1994;28(2),30-41.

Regarding revisions required by the reviewer #2:

1. The abstract provides a good summary. State explicitly how university administrators can use these findings in real crisis situations. Provide a sentence on how universities can incorporate these new strategies into their crisis response plans.

Firstly, we would like to express our sincere gratitude to the reviewer for acknowledging and recognizing our work. The findings of this study can indeed enable university administrators to apply these new strategies in crisis response, which can be incorporated into their crisis response plans. Therefore, based on your suggestion, we have added a sentence in the abstract to convey this meaning. The revised sentence is as follows:

“The findings provided guiding significance for Chinese universities administrators in developing and implementing effective crisis response strategies in real-life crisis situations, enabling them to adjust and optimize their plans by incorporating innovative strategies to enhance the effectiveness of responses.”

In addition, we want to explain to you that, for your convenience during the review process, we originally intended to highlight the revisions made. However, according to the journal editor's instruction, the main manuscript file should not contain any tracked changes or highlighting, as this will be used in the production process if the manuscript is accepted. Therefore, we will endeavor to explain all the revisions in the response letter instead.

2. The paper justifies excluding third-party narratives, but it does not fully address why stakeholder perceptions were not considered. Since crisis communication is about perception management, future research should examine how students, faculty, and the public perceive these responses.

Thank you for your comment. The reviewers have provided insightful observations on the core issues of this study. Indeed, we are well aware that the essence of crisis communication goes beyond the unilateral strategic choices of information issuers. Crisis response is a two-way process of perception management, which hinges on whether stakeholders (students, faculty, the public, etc.) truly receive, understand, and accept these strategies. There may be slight deviations between official strategies and stakeholders' perceptions. More importantly, as we have stated in the article, due to the fact that stakeholders' perceptions may be influenced by their own interests, as well as potential emotional factors and biases towards the organization, it is not conducive to maintaining the objectivity of this study.

However, the exclusion of third-party narratives and even stakeholder perspectives in this study is not in opposition to the reviewers' viewpoints, rather, it reflects a different emphasis in research focus. This study focuses on the coding of institutionalized strategies in universities' official responses, and reveals what response strategies Chinese universities choose under the institutional logic of organizational legitimacy and policy compliance when responding to crises. This methodological design does not neglect stakeholders' perceptions but rather delineated them as distinct phases, firstly, we focused on analyzing which strategies the university as the responding entity has employed in the official documents. In the future, based on your suggestions, the stakeholders’ perceptions will be further taken into consideration, representing another independent analytical layer that requires different methodological approaches.

The strategies identified in this study initially demonstrate the strategies applied by universities as response entities in their crisis responses. Official documents, as authoritative sources of institutional intent, enable us to track how universities make strategic choices within recognized discourse frameworks in response to crises. By focusing on universities' official institutional response strategies, it is crucial for identifying systematic patterns rather than individual cases. The research findings can serve as a baseline framework, against which future related studies can measure the degree of alignment or inconsistency in cognitive outcomes. This study does not deny the importance of perception management but rather lays the foundation for the future research.

According to your comments, we have made two changes to the article, one is ROW251-261 about the description of excluding third-party narratives and stakeholder perceptions, and the other is ROW609-616 about future research direction at the end of the article.

“As the purpose of this research is to determine how the Chinese universities communicated about crises in their external communications, news articles or interviews of school staff were excluded to avoid the subjectivity of a third party in the reporting and framing of the universities’ responses. All documentary evidence is officially released by universities. Meanwhile, due to stakeholders perceptions may be influenced by their own interests, and potential emotional factors and biases towards the organization, in order to focus on the coding of institutionalized strategies in universities' official responses, and reveal what response strategies Chinese universities choose under the institutional logic of organizational legitimacy and policy compliance when responding to crises, while maintaining the objectivity of the research, the influence of stakeholders perceptions are also excluded in this study.”

“Furthermore, the strategies identified in this study initially demonstrate the strategies applied by universities as response entities in their crisis responses. However, the essence of crisis communication goes beyond the unilateral strategic choices of information issuers. Crisis response is a two-way process of perception management, which hinges on whether stakeholders (students, faculty, the public, etc.) truly receive, understand, and accept these strategies. In the future, the stakeholders’ perceptions will be further taken into consideration, the findings of this research can serve as a baseline framework, against which future related studies can measure the degree of alignment or inconsistency in cognitive outcomes.”

3. The newly identified strategies are mapped onto existing SCCT categories, but some seem to be stand-alone approaches rather than fitting neatly into SCCT. Instead of force-fitting into SCCT, consider discussing how these strategies might complement or challenge the existing framework.

We sincerely appreciate the insightful observations made by the reviewers regarding the relationship between our research findings and the SCCT framework. We believe that the reviewers' opinions might suggest that mapping the new strategies onto the existing framework of SCCT could potentially obscure its unique implications, especially in specific cultural contexts, thereby missing opportunities to challenge or expand the theory. We consider your opinion to be highly innovative and it has greatly inspired and enlightened us. However, after our in-depth discussion, we believe that making further changes to the core framework of this study in the fourth round of minor revision seems a little bit risky. Moreover, incorporating the newly discovered strategies into the SCCT framework is not entirely without basis. This is based on the previous research results of other scholars (as we mentioned in the previous response letter and the article), and secondly, our conclusion is reached through the consensus of multiple rounds of coding and discussions by the coders. We admit that the current framework is not perfect, and it is not immutable either, but if the framework is modified, it might require the coders to have a deeper discussion and potentially overturn and rewrite the main part of this article. Nevertheless, your suggestions have greatly inspired us. We will deeply reflect on your suggestions and hope to achieve further innovation in subsequent research. Thank you for your professional advice and we hope that you can accept and understand our explanation for this matter.

4. Condense the discussion of response strategies per crisis cluster to avoid redundancy.

Thanks to the reviewer's suggestion. According to your suggestion, firstly, we reorganize the discourse of the discussion about applicable response strategies for different crisis clusters in section 4.4 by streamlining the expression without reducing the content, and consider the sentence changes to enhance readability. Secondly, we simplified and modified the redundant content in the conclusion paragraph of section 4.4, that is, the last paragraph of section 4.4. The revised contents are as follows:

“4.4 Applicable response strategies for different crisis clusters

SCCT emphasizes the adaptation of crisis response strategies to different situations. This study conducts a comparative analysis of the use of crisis response strategies in different situations. As Fig 2 shows, in the intentional cluster, the primary strategy of rebuild, including compensation, apology, commitment, and remedy, are most frequently employed. Chinese universities tend to use accommodative strategies when responsibility attribution is strong. Accountability and punishment strategies are also common, reflecting a similar accommodative approach. Endorsement, concern, and condolence are the most used secondary strategies.

For the accident cluster, the strategy of diminish, including excuse, transference, and separation are prevalent. These strategies are optimal in situations with moderate responsibility attribution. Additionally, explanation is a key primary strategy, and reminder and warning are common secondary strategies which are well-suited to crises in the accident cluster.

In the victim cluster, the primary strategy of deny, which includes attack the accuser, denial, and scapegoat, are widely used. Although the public might not hold the universities directly responsible for the crisis in this situation, restoring public trust depends on the universities’ response. Deny is effective in stabilizing public emotions with minimal cost in situations with weak responsibility attribution. Appeal, valued highly, gratitude, and statement are secondary strategies frequently employed. Notably, investigation and information disclosure strategies were applicable across all crisis situations.

Therefore, the results of this study prove that the actual situation of crisis response in Chinese universities is consistent with the concept of SCCT, that is, the degree of accommodation of applicable strategies increases with the degree of responsibility. The conclusion of “the rebuild strategies are the safest crisis response for the reason of rebuild strategies addressing victims well” is not universally applicable, because using overly accommodating strategies in the inappropriate situations makes the cost higher to the organization but may not see any increased yield in reputation protection. Therefore, the effectiveness of the response to the crisis is not due to the unlimited accommodative response strategy, but to the adaptation of the response strategy to the category of the crisis. The results of this study provide a summary of recommendations for the use of crisis response strategies in Chinese universities.”

5. Table 5 requires restructuring for clarity. Consider sentence variation to improve readability.

Thanks to reviewer for your professional advice. Actually, during the first round of revision, Reviewer #1 deemed there to be too many tables in the manuscript and suggested removing some of them, while describing the contents of the tables in simple text without losing any details. Therefore, in the first round of revision, we have already removed Table 5 and summarized it in a simple way. We are not yet certain if a minor misunderstanding within the review system has led to a discrepancy in information. To address this, we have carefully examined the revised manuscript we submitted, ensuring that there is no Table 5 included. We have also checked the files in the submission system and confirmed that Table 5 is not present. However, we have modified the relevant content to improve readability by increasing sentence variability. See the answer in comment 4 for details. We remain grateful for your review of our manuscript, and we will make every effort to address any other revisable comments.

We sincerely thank you again for your meticulous review of our research and your valuable comments. Your feedback is of great importance to us and has prompted us to make corresponding revisions and improvements to the paper. We are confident that these enhancements have enhanced the quality and clarity of the paper. We sincerely hope that after careful revisions, the manuscript will meet your approval and gain the opportunity for publication. Thank you once again for your hard work and professional guidance.

---

## [Decision Letter · Decision Letter 4]

12 Mar 2025

Choosing the right strategies:An analysis of crisis response strategies in Chinese universities

PONE-D-24-14141R4

Dear Dr. Yongshi,

We’re pleased to inform you that your manuscript has been judged scientifically suitable for publication and will be formally accepted for publication once it meets all outstanding technical requirements.

Kind regards,

Ahmed Meri, Ph.D.

Academic Editor

PLOS ONE

Reviewers' comments:

Reviewer's Responses to Questions

**Comments to the Author**

1. If the authors have adequately addressed your comments raised in a previous round of review and you feel that this manuscript is now acceptable for publication.

Reviewer #2: All comments have been addressed

2. Is the manuscript technically sound, and do the data support the conclusions?

Reviewer #2: Yes

3. Has the statistical analysis been performed appropriately and rigorously? 

Reviewer #2: Yes

4. Have the authors made all data underlying the findings in their manuscript fully available?

Reviewer #2: Yes

5. Is the manuscript presented in an intelligible fashion and written in standard English?

Reviewer #2: Yes

6. Review Comments to the Author

Reviewer #2: The study is a valuable addition to crisis communication literature, especially in the Chinese university context. The expansion of Situational Crisis Communication Theory (SCCT) with new strategies is noteworthy.

While the study identifies 17 new strategies beyond SCCT, the rationale behind their classification and uniqueness could be strengthened. How do these differ significantly from existing SCCT strategies?

The authors acknowledge the limitation of excluding stakeholder perceptions. However, a brief discussion on how these perceptions might influence strategy effectiveness would enhance the paper.

The discussion of response strategies per crisis cluster is somewhat repetitive. A more condensed version would improve readability.

Ensure that all references are complete and formatted correctly.

7. PLOS authors have the option to publish the peer review history of their article (what does this mean? ). If published, this will include your full peer review and any attached files.

**Do you want your identity to be public for this peer review?** For information about this choice, including consent withdrawal, please see our Privacy Policy .

Reviewer #2: No

---

## [Editor Report · Acceptance letter]

PONE-D-24-14141R4

PLOS ONE

Dear Dr. Yongshi,

I'm pleased to inform you that your manuscript has been deemed suitable for publication in PLOS ONE. Congratulations! Your manuscript is now being handed over to our production team.

Kind regards,

on behalf of

Dr. Ahmed Meri

Academic Editor

PLOS ONE